# Exploiting the Kurtosis Concentration Property for Image quality improvement

## Abstract

Diffusion models have significantly advanced generative AI in terms of creating and editing naturalistic images. However, improving the image quality of generated images is still of paramount interest. In this context, we propose a generic kurtosis concentration (KC) loss, which can be readily applied to any standard diffusion model pipeline to improve image quality. Our motivation stems from the *projected kurtosis concentration property* of natural images, which states that natural images have nearly constant kurtosis values across different band-pass versions of the image. To improve the image quality of generated images, we reduce the gap between the highest and lowest kurtosis values across the band-pass versions (e.g., Discrete Wavelet Transform (DWT)) of images. In addition, we also propose a novel condition-agnostic perceptual guidance strategy during inference to further improve the image quality. We validate the proposed approach for three diverse tasks, viz., (1) personalized few-shot finetuning using text guidance, (2) unconditional image generation, and (3) image super-resolution. Integrating the proposed KC loss and perceptual guidance has improved the perceptual quality across all these tasks in terms of FID, MUSIQ score, and user evaluation. Code and README are provided in the supplementary material.

## 1 Introduction

Multi-modal generative AI has advanced by leaps and bounds with the advent of the diffusion model. Large-scale text-to-image diffusion models, e.g., DALLE Ramesh et al. (2022), Stable-diffusion Rombach et al. (2022) synthesize high-quality images in diverse scenes, views, and lighting conditions from text prompts. These models generate high-quality and diverse images since they have been trained on a large collection of image-text pairs and can capture the visual-semantic correspondence effectively. While diffusion models generate images that appear highly realistic, recent studies have demonstrated that these images can still be distinguished from natural ones using advanced image forensic tools Corvi et al. (2023). This suggests that although state-of-the-art generative models excel at tasks like image editing, they often leave behind subtle, unnatural artifacts. Ensuring high image quality is therefore critical for various generative tasks, such as personalized few-shot finetuning Ruiz et al. (2022); Kumari et al. (2022), super-resolution Karras et al. (2022); Dhariwal & Nichol (2021), image restoration, and unconditional image generation.

Our goal is to improve the image quality using natural image statistics by exploiting the well-known kurtosis concentration property of natural images Zhang & Lyu (2014); Zoran & Weiss (2009); Wainwright & Simoncelli (1999). This property states that natural images have nearly constant kurtosis (fourth order moment) values across different band-pass (e.g., Discrete Cosine Transform (DCT), Discrete Wavelet Transform (DWT)) versions of the images Zhang & Lyu (2014). Inspired by this property, we propose a novel kurtosis concentration (KC) loss, which is generic and applicable to any diffusion-based pipeline. More specifically, this loss minimizes the gap in the kurtosis of an image across band-pass filtered versions and improves the quality of the generated images. We also propose a novel perceptual guidance (PG) strategy during inference which is agnostic to conditioning (e.g., text/class) and further improves image quality. Both KC loss and PG strategies are general-purpose and do not require any labels. It can be adapted to various generative tasks with minimal effort. In this work, we experiment with diverse tasks: (1) personalized few-shot finetuning of text-to-image diffusion model, (2) unconditional image generation, and (3) image super-resolution.

Our major contributions are as follows:

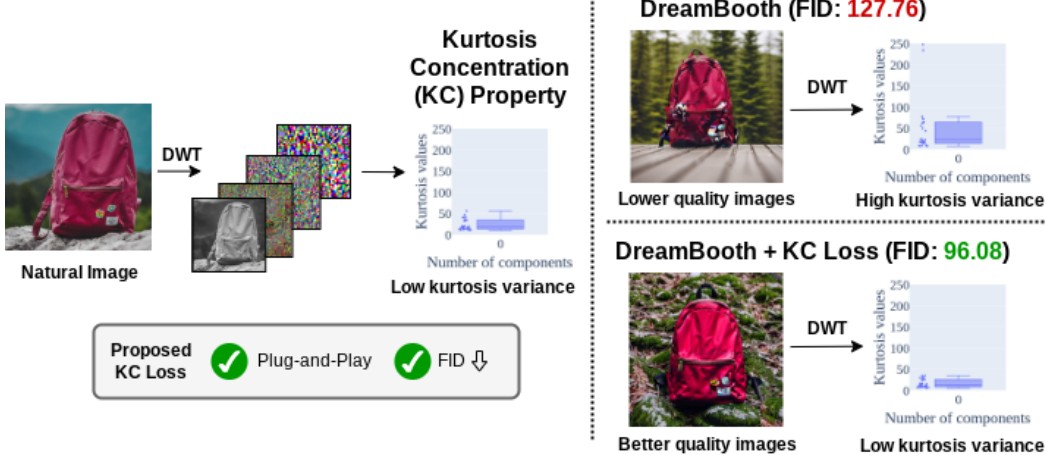

Figure 1: Overview of DiffNat. We utilize the kurtosis concentration property of natural images, which states the kurtosis values across different bandpass filtered (Discrete Wavelet Transform) versions of the images tend to be constant. As can be observed in this figure, the 50 percentile of the kurtosis values resides in the blue box, which indicates the concentration of the kurtosis values. In the left fig., for natural images, this spread is relatively smaller. In the top right fig., Kurtosis spread appears to be higher in diffusion-generated images by DreamBooth Ruiz et al. (2022), resulting in lower quality images. The figure in the bottom right shows that the addition of KC loss improves image quality in terms of FID and reduces the kurtosis variance.

- We introduce DiffNat - a framework for improving the image quality of diffusion models using the kurtosis concentration property of natural images. To the best of our knowledge, we are the first to propose this loss based on natural image statistics.

- We provide insights on how reducing kurtosis improves image quality. This is the primary motivation for the proposed loss function.

- A novel condition-agnostic perceptual guidance strategy is proposed which further improves image quality.

- We validate the proposed KC loss and PG strategy on diverse generative tasks, e.g., (1) personalized few-shot finetuning of text-to-image diffusion model using text guidance, (2) unconditional image generation, and (3) image super-resolution. Experiments indicate that incorporating the proposed KC loss and PG enhances perceptual quality across various tasks and benchmarks, and this improvement has been validated through a user study.

## 2 RELATED WORK

**Generative models.** Recent progress in generating high-fidelity, diverse images from text inputs has been remarkable. Initially, GAN-based methods dominated text-to-image generation Qiao et al. (2019); Tao et al. (2022); Liao et al. (2022); Zhu et al. (2019); Ruan et al. (2021), but recent advances have shifted towards diffusion models like Stable Diffusion Rombach et al. (2022) and Imagen Saharia et al. (2022), which leverage large datasets for training. Text-based image editing has also advanced significantly; GAN-based approaches have improved with CLIP Radford et al. (2021), while diffusion-based methods offer better control and impressive results Ruiz et al. (2022); Kumari et al. (2022); Gal et al. (2022). Personalization techniques such as Textual Inversion Gal et al. (2022), DreamBooth Ruiz et al. (2022), and Custom Diffusion Kumari et al. (2022) allow for the creation of unique images by embedding subjects or concepts into the model's output. In unconditional image generation, the Denoised Diffusion Probabilistic Model (DDPM) Ho et al. (2020) is a leading method, providing superior image quality through variational inference and image-space denoising. For conditional tasks like image super-resolution, guided diffusion Dhariwal & Nichol (2021) and latent diffusion models Karras et al. (2022) are highly effective, producing high-resolution images from low-resolution inputs.

**Natural Image Statistics.** Natural images have interesting scale-invariance and noise properties Zoran & Weiss (2009), which have been used for image restoration problems. Projected KC property of natural images, i.e., natural images tend to have constant kurtosis values across different band-pass (DCT, DWT) filtered versions has been used for blind forgery detection Zhang & Lyu (2014). In-

spired by these observations, we propose a novel loss function based on natural image statistics for generating better quality images.

# 3 METHOD

In this section, we present the concept of KC loss (Sec. 3.2) which can be applied to various generative tasks for enhancing the quality of generated images. We start by providing a basic understanding of the KC property of natural images.

## 3.1 KURTOSIS CONCENTRATION PROPERTY

**Definition 1** *Kurtosis: It is a measure of the "peakedness" of the probability distribution of a random variable Zhang & Lyu (2014). For a random variable $x$, its kurtosis is defined as,*

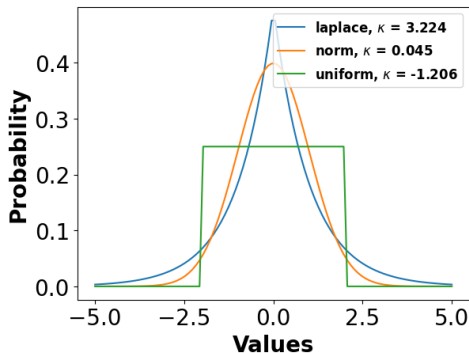

Figure 2: Kurtosis of various distributions where kurtosis captures the peakedness of distributions.

$$\kappa(x) = \frac{\mu_4(x)}{(\sigma^2(x))^2} - 3. \tag{1}$$

*where $\sigma^2(x) = \mathbb{E}_x[(x - \mathbb{E}_x(x))^2]$ and $\mu_4(x) = \mathbb{E}_x[(x - \mathbb{E}_x(x))^4]$ are the second order and fourth order moment of $x$. For example, the Gaussian random variable has a kurtosis value of 0.*

Kurtosis of well-known distributions is shown in Fig. 2. We can observe that a positive kurtosis indicates that the distribution is more peaked than the normal distribution and negative kurtosis indicates it to be less peaked than normal distribution Zhang & Lyu (2014). Kurtosis is a useful statistic used for blind source separation Naik et al. (2014) and independent component analysis (ICA) Stone (2002).

For a random vector $x$, we define the kurtosis of the 1D projection of $x$ onto a unit vector $w$ as a projection kurtosis, i.e., $\kappa(w^T x)$. Projection kurtosis is an effective measure of the statistical properties of high-dimensional data. E.g., if $x$ is a Gaussian, its projection over any $w$ has a 1D Gaussian distribution. Therefore, its projection kurtosis is always zero, which exhibits the kurtosis concentration (to a single value, i.e., zero) of Gaussian.

It is well-known that natural images can be modeled using zero-mean Gaussian Scale Mixture (GSM) vector Zoran & Weiss (2009); Zhang & Lyu (2014); Lyu et al. (2014); Wainwright & Simoncelli (1999). Next, we analyze an interesting property of the GSM vector.

**Lemma 1** *A GSM vector $x$ with zero mean has the following probability density function:*

$$p(x) = \int_0^\infty \mathcal{N}(x; 0, z\Sigma_x) p_z(z) dz \tag{2}$$

*and its projection kurtosis is constant with respect to the projection direction w, i.e.,*

$$\kappa(w^T x) = \frac{3 var_z\{z\}}{\mathcal{E}_z\{z\}^2} \tag{3}$$

*where $\mathcal{N}(x; 0, z\Sigma_x)$ denotes a Gaussian distribution with zero mean and covariance matrix $z\Sigma_x$, with $z$ a positive random variable with density $p_z(z)$. $\mathcal{E}_z\{z\}$ and $var_z\{z\}$ are the mean and variance of latent variable $z$ respectively.*

*Proof.* The proof is provided in the Appendix.

This result by Zhang & Lyu (2014) shows that projection kurtosis is constant across projection directions (e.g., wavelet basis), which provides theoretical insights of the kurtosis concentration property, which we will discuss next.

**Kurtosis Concentration Property**: It has been observed that for natural images, projection kurtosis values across different band-pass filter channels tend to be close to a constant value. This is termed as kurtosis concentration property of natural images Zhang & Lyu (2014); Zoran & Weiss (2009); Lyu

et al. (2014); Bethge (2006); Lyu & Simoncelli (2009); Wainwright & Simoncelli (1999). It can also be interpreted as an implication of Lemma 1, if we consider patches of natural images as zero-mean GSM vector ( Zoran & Weiss (2009), Wainwright & Simoncelli (1999)) and projection directions correspond to band-pass filters, e.g., DWT.

One intuitive reasoning of the projected kurtosis concentration property is given as follows. It is observed that the distribution ($p(x, \alpha, \beta)$) of different bandpass (DWT) filtered versions of natural images follows a generalized Gaussian density of the form Zhang & Lyu (2014); Zoran & Weiss (2009).

$$p(x, \alpha, \beta) = \frac{\beta}{2\alpha\Gamma(1/\beta)} exp(-\frac{|x|}{\alpha})^{\beta} \tag{4}$$

where $\alpha$, $\beta$ are scaling parameters and $\Gamma(.)$ is the Gamma function. The kurtosis of this function is given by Zoran & Weiss (2009),

$$\kappa = \frac{\Gamma(1/\beta)\Gamma(5/\beta)}{\Gamma(3/\beta)^2} \tag{5}$$

Empirically, it has been shown that for natural images, $\beta$ is relatively small values ranges from 0.5 to 1 Zoran & Weiss (2009), and this kurtosis value tend to be constant Zhang & Lyu (2014); Zoran & Weiss (2009); Lyu et al. (2014); Wainwright & Simoncelli (1999), independent of $\alpha$ or $x$.

We investigate and experimentally verify this property for natural images on large datasets, e.g., FFHQ dataset (Fig. 8(c)), Dreambooth dataset (Fig. 14(c), Appendix), Oxford-flowers dataset (Fig. 15(c), Appendix). We conclude that this property actually holds for both object datasets (Dreambooth dataset, Oxford flowers), face dataset (FFHQ) with sufficient variations in viewpoint, scale, illumination, color, objects, pose, lighting condition etc. Analysis of kurtosis difference has been shown in Fig. 8 and Fig. 14, Fig. 15 (Appendix), which clearly shows that the difference of kurtosis values are higher in diffusion generated images compared to natural images in these datasets.

Next, we establish the relation between the projection kurtosis of the noisy version of the image and the corresponding signal-to-noise ratio.

**Lemma 2** *If the noisy version of the natural image is denoted by, y = x + n, where x is a whitened GSM vector (normalized natural image) and n is a zero-mean white Gaussian noise with variance $\sigma^2 I$, x and n are mutually independent of each other, then the projection kurtosis of y, $\kappa(w^T y)$ can be expressed as:*

$$\kappa(w^T y) = \kappa(w^T x)\Big(1 - \frac{c}{SNR(y)}\Big)^2 = \frac{3var_z\{z\}}{\mathcal{E}_z\{z\}^2}\Big(1 - \frac{c}{SNR(y)}\Big)^2 \tag{6}$$

*where Signal-to-Noise Ratio (SNR) is defined as, $SNR(y) = \frac{\sigma^2(y)}{\sigma^2(n)}$ and c is a constant.*

*Proof.* The proof is provided in the Appendix.

This result utilizes the fact that natural images have constant projection kurtosis, as stated in Lemma 1. Next, we connect projection kurtosis minimization to denoising.

**Proposition 1** *Minimizing projection kurtosis denoise input signal.*

From Lemma 2, we can observe there exists an inverse relation between the projection kurtosis and image quality (SNR), therefore minimizing projection kurtosis will increase SNR and the image will be denoised better.

The primary objective of diffusion models is to learn denoising from a noisy image or latent embedding in order to generate a clean image. Then by Lemma 2, the projection kurtosis minimization results in the denoising (high SNR) of the reconstructed image. In the case of diffusion models, the underlying denoising UNet is trained using mean squared error objective w.r.t the reconstructed image and the clean image. During inference, the reconstructed image is iteratively denoised and refined for $T$ steps to generate the final image with higher quality. Therefore, adding an objective to minimize the projection kurtosis of the reconstructed image, i.e., increasing the SNR (Lemma 2) would effectively lead to better denoising at each step and the final image would be of improved quality. Thus, the proposed KC loss serves as an additional regularizer for training the denoising neural network, without compromising its underlying theoretical framework (Appendix Sec. D).

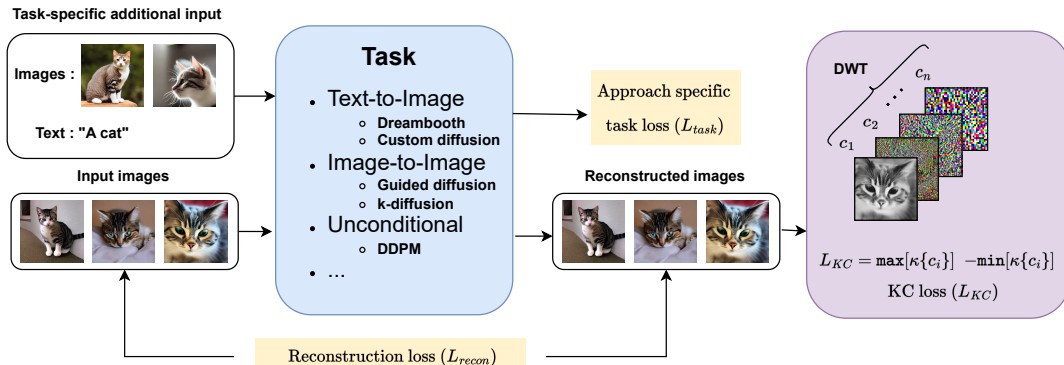

Figure 3: Overview of DiffNat. The proposed kurtosis concentration loss can be integrated into any diffusion-based approach for various tasks (e.g., text-to-image generation (DreamBooth, Custom diffusion), super-resolution image-to-image generation (Guided diffusion, k-diffusion), unconditional image generation (DDPM)). In addition to the task-specific losses, and general reconstruction loss, we incorporate the kurtosis concentration loss ($L_{KC}$), which operates on the reconstructed images and minimizes the kurtosis deviation (i.e., $\max[\kappa\{c_i\}]$ - $\min[\kappa\{c_i\}]$) across Discrete Wavelet Transform (DWT) filtered version of the reconstructed image, Here, $c_1$, $c_2$ .. are DWT filtered version of the reconstructed image and $\kappa(x)$ denote kurtosis of $x$.

## 3.2 KURTOSIS CONCENTRATION (KC) LOSS

In this work, we introduce a novel KC loss function for training deep generative models, leveraging the KC property of natural images to improve perceptual quality. Unlike previous approaches Zhang & Lyu (2014) that used the KC property for tasks like noise estimation and source separation, our KC loss can be integrated into any diffusion pipeline, and we validate its effectiveness with state-of-the-art diffusion models.

Suppose, we need to train or finetune a diffusion model $f_\theta$ using input training images ($x$) with or without a conditioning vector $c$. The conditioning vector could be text, image, or none (in the case of the unconditional diffusion model). Given an initial noise map $\epsilon \sim N(0, I)$, and a conditioning vector $c$, the generated images obtained from $f_\theta$ is given by $x_{gen} = f_\theta(x, \epsilon, c)$. Typically, the diffusion model is trained to minimize the $l2$ distance between the ground truth image ($x$) and the noisy image ($x_{gen}$) Dhariwal & Nichol (2021) or their corresponding latent in case of Latent Diffusion Model (LDM) Rombach et al. (2022). Without loss of generality, we are referring to that as reconstruction loss ($L_{recon}$) between the ground-truth image ($x$) and the generated image ($x_{gen}$), denoted by,

$$L_{recon} = \mathbb{E}_{x,c,\epsilon}[\, ||x_{gen} - x||_2^2] \tag{7}$$

Note that, for diffusion models trained to predict the added noise, we could deterministically obtain the intermediate clean image from the predicted noise and apply the loss to that. Next, we will describe the KC loss. Note, that the KC property holds across different bandpass transformed domains (DCT, DWT, fastICA) and we choose DWT because it is widely used due to its hierarchical structure and energy compaction properties E Woods & C Gonzalez (2008). Typically, DWT transforms images into LL (low-low), LH (low-high), HL (high-low), HH (high-high) frequency bands and each of the sub-bands contains several sparse details of the image. E.g., LL and HH subband contain a low-pass and high-pass filtered version of the image respectively Zhang & Lyu (2014). The generated image $x_{gen}$ is then transformed using Discrete Wavelet Transform (DWT) with kernels $k_1, k_2, .., k_n$ producing filtered images $g_{gen,1}, g_{gen,2}, .., g_{gen,n}$ respectively, such that, $g_{gen,i} = F_{k_i}(x_{gen})$. Here, $F_l$ denotes the discrete wavelet transform with kernel $l$.

Now, kurtosis values of these $g_{gen,i}$ should be constant by the KC property, therefore, we minimize the difference between the maximum and minimum values of the kurtosis of $g_{gen,i}$'s to finetune the model using the loss,

$$L_{KC} = \mathbb{E}_{x,c,\epsilon}[\max(\kappa\{g_{gen,i}\}) - \min(\kappa\{g_{gen,i}\})] \tag{8}$$

Here, $\kappa(x)$ is kurtosis of $x$. Note that, this loss is quite generic and can be applied to both image or latent diffusion models for training. In the case of latent diffusion models, we need to transform the latent to image space (via a pretrained VQVAE), before applying this loss, since this prior holds

for image space only. In case of applying this loss to any task $T$ (DreamBooth, super-resolution, unconditional image generation), the overall loss ($L$) function would be, $L = L_{task} + L_{recon} + L_{KC}$, where $L_{task}$ is the task-specific loss.

**Lemma 3** *KC loss is differentiable and Lipschitz continuous with Lipschitz constant 2.*

*Proof.* The proof is provided in the Appendix.

We are taking maximum and minimum across kurtosis values, therefore KC loss is differentiable and so is the combined loss, since differentiability preserves over addition. The notion that KC loss is Lipchitz continuous with Lipschitz constant of value 2, indicates that the changes in loss will be bounded by the changes in input. This implies that the model is less sensitive to small perturbations in the input, leading to more stable predictions. We have experimentally validated this in Sec. 5.

### 3.3 PERCEPTUAL GUIDANCE (PG) DURING INFERENCE

Directly applying the KC loss to diffusion training improves sample quality over vanilla diffusion. Additionally, we introduce a novel PG mechanism during inference to further enhance perceptual quality. Intuitively, this is similar to classifier-free guidance, where the diffusion model is run both conditionally and unconditionally at every step, and the difference in the output is considered as the gradient direction towards the condition. While classifier-free guidance can improve the perceptual quality as a by-product, it has several potential drawbacks, such as: (1) It works with conditional models only. (2) The goal of classifier-free guidance is to generate images aligned with the condition, which may not explicitly align with the perceptual quality. Instead, we propose to produce two samples of varying perceptual quality (with two different models with and w/o KC loss) and amplify the intermediate output at each step toward better perceptual quality. The key advantages of such a design are: (1) it is independent of condition, (2) can operate in parallel to traditional classifier-free guidance.

Suppose a baseline diffusion model ($f_B$) is trained with $L_{task} + L_{recon}$. We also train another model ($f_P$) with the proposed KC loss, i.e., $L_{task} + L_{recon} + L_{KC}$. During inference, let $\epsilon_t$ be the noisy latent at time step $t$. The output of the two diffusion model $f_B$ and $f_P$ would be $\epsilon^*_{t,B} = f_B(\epsilon_t, t, c)$, and $\epsilon^*_{t,P} = f_P(\epsilon_t, t, c)$, respectively, where $c$ is the conditional vector, which can be null as well. We calculate the latent for the next time step $t - 1$ as $\epsilon_{t-1} = \epsilon^*_{t,B} + \lambda(\epsilon^*_{t,P} - \epsilon^*_{t,B})$. This will be iterated $T$ times to generate the final sample.

After a simple algebric manipulation, the equation becomes, $\epsilon_{t-1} = \lambda * \epsilon^*_{t,P} - (\lambda - 1)\epsilon^*_{t,B}$. Thus, if $\lambda$ is greater than 1, we essentially substract the part corresponding to lower perceptual quality (baseline, no KC loss) from a higher quality one (perceptual model). Intuitively, this is similar to the vector direction from point B to point A, where B is baseline model and A is perceptual model, i.e., direction towards a better perceptual quality in the diffusion noisy latent space.

To prevent numerical issues like overflowing when linearly combining scaled output of two different models, we use the standard clipping strategy used in guided diffusion Dhariwal & Nichol (2021), stable diffusion Rombach et al. (2022) etc. For example, we keep the variance of the model fixed during training and inference. We only calculate the mean during inference using the output of the pre-trained diffusion UNets $f_B$ and $f_P$. After performing a linear combination, we clamp the final value between [-1, +1] to preserve the scale of the distribution during reverse diffusion.

Note that, the PG steers the generated images to gradually move towards better perceptual quality, with or without any conditional guidance. The PG can also be interpreted as a special form of conditional guidance, where the same input is conditioned on two models, one trained vanilla, and another trained with perceptual KC loss, and the difference between their outputs guides the denoising process during inference, leading to improved perceptual quality. We experimentally verify the efficacy of perceptual guidance.

### 3.4 INTUITIVE JUSTIFICATION

**Why diffusion generated images have higher kurtosis values?** Natural images typically exhibit smooth transitions and structured patterns, leading to a pixel intensity distribution with fewer outliers and, consequently, lower kurtosis Zoran & Weiss (2009); Lyu et al. (2014); Wainwright & Simoncelli

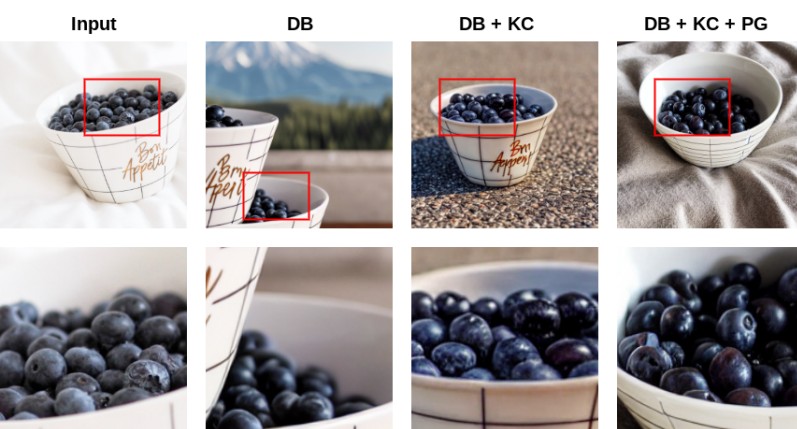

Figure 4: Qualitative comparison of with/without KC loss and PG in DreamBooth (DB). The bottom image (with KC loss) shows better image quality and reflections on the bowl full of berries (best viewed in color).

(1999); Bethge (2006). In contrast, diffusion models generate images through iterative refinement of pure noise into coherent structures. Due to imperfections in the trained UNet and finite denoising steps, residual high-frequency noise may persist in the final output, leading to more extreme pixel intensity values and contributing to heavier tails in the distribution Zhang et al. (2023). This phenomenon is more effectively characterized in the frequency domain (wavelet transform), which generally correspond to higher kurtosis values Zhang & Lyu (2014) (Fig. 8).

**How does KC loss improve image quality?** Intuitively, minimizing the KC loss can be seen as a locality-aware smoothing of the data distribution to enhance perceptual quality. For instance, in Fig. 6, the output of GD without KC loss exhibits undesirable abrupt changes near the eye region. Ideally, we should first detect such regions and then apply suitable operations to improve perceptual quality. However, performing a spatially varying refinement is challenging, and globally applying such filters might be sub-optimal for other regions. It has been observed that the wavelet coefficients of natural images follow a Generalized Gaussian density Moulin & Liu (1999); Sharifi & Leon-Garcia (1995); Mallat (1989), and Kurtosis quantifies the heaviness of the tails and peakedness of a distribution compared to the Gaussian distribution Zhang & Lyu (2014); Maier (2021). Therefore, kurtosis across bandpass wavelet filtered versions of the image automatically provides locality of the abrupt changes and minimizing KC loss performs locality-aware smoothing of the data distribution. In Fig. 6, incorporating KC loss enhances the generated eye region.

## 4 EXPERIMENTS

We evaluate the efficacy of the proposed loss for three tasks - (1) personalized few-shot finetuning of diffusion model using text guidance, (2) unconditional image generation, and (3) image super-resolution.

### 4.1 TASK 1: PERSONALIZED FEW-SHOT FINETUNING USING TEXT GUIDANCE

In this section, we address the problem of finetuning the text-to-image diffusion model from a few examples for text-guided image generation in a subject-driven manner. Specifically, given only a few images (e.g., 3-5) of a particular subject without any textual description, our task is to learn the subject-specific details and generate new images of that particular subject in different conditions specified by the text prompt.

To evaluate the efficacy of KC loss in this task, we build upon two popular methods, (1) Dream-Booth Ruiz et al. (2022), and (2) Custom diffusion Kumari et al. (2022). We evaluate both approaches with/without KC loss on the DreamBooth dataset for a fair comparison. When adding the proposed KC loss to these approaches, we obtain performance improvements in visual quality, i.e., FID Lucic et al. (2018), MUSIQ score Ke et al. (2021) and subject and prompt fidelity metrics as well (DINO, CLIP-I, CLIP-T) as shown in Tab. 1. The qualitative results are shown in Fig. 4. We follow the same setup for the DreamBooth and Custom diffusion baselines. We have also compared with another naturalness loss, i.e., LPIPS loss Zhang et al. (2018) as a baseline. Proposed PG further improves perceptual quality. Additionally, for KC loss, we decompose the reconstructed images using 27

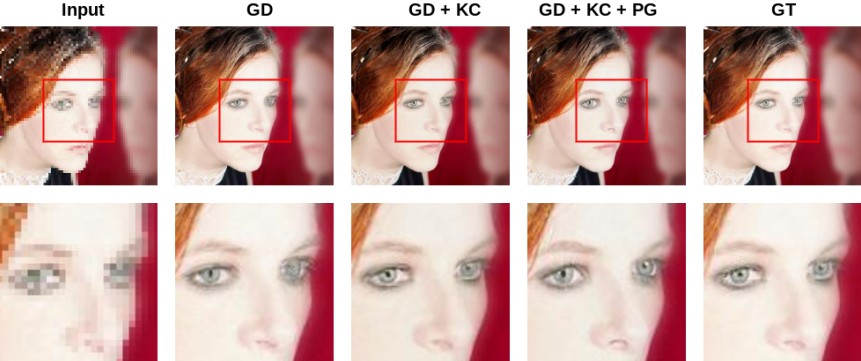

Figure 5: Comparison of unconditional image generation (DDPM) with/without KC loss and PG. Integrating KC loss and PG significantly improves image quality, whereas DDPM-generated images have unnatural image artifacts.

Figure 6: Qualitative comparison of with/without KC loss in guided diffusion (GD). The bottom image (with KC loss) has better eye and hair details (best viewed in color).

'Daubechies' filter banks and get the average difference of the kurtosis values as a loss function. More training details and ablation are provided in the Appendix.

**Human evaluation.** Since perceptual metrics are not always reliable, we also conducted a human preference study using Amazon Mechanical Turk (AMT) for (1) subject fidelity assessment and (2) image quality ranking. For the subject fidelity assessment, we evaluated the visual similarity of real and generated images, both with and without KC loss, to the actual subject. We asked around 5000 visual similar-

Table 1: Comparison of personalized few-shot finetuning task

| Method | Image quality | | Subject fidelity | | Prompt fidelity |
|---|---|---|---|---|---|
| | FID ↓ | MUSIQ ↑ | DINO ↑ | CLIP-I ↑ | CLIP-T ↑ |
| DB Ruiz et al. (2022) | 111.76 | 68.31 | 0.65 | 0.81 | 0.31 |
| DB Ruiz et al. (2022) + LPIPS | 108.23 | 68.39 | 0.65 | 0.80 | 0.32 |
| DB + KC loss (Ours) | 100.08 | 69.78 | 0.68 | 0.84 | 0.34 |
| DB + KC loss + PG (Ours) | **93.45** | **70.82** | **0.70** | **0.86** | **0.35** |
| CD Kumari et al. (2022) | 84.65 | 70.15 | 0.71 | 0.87 | 0.38 |
| CD Kumari et al. (2022) + LPIPS | 80.12 | 70.56 | 0.71 | 0.87 | 0.37 |
| CD + KC loss (Ours) | 75.68 | 72.22 | 0.73 | 0.88 | 0.40 |
| CD + KC loss + PG (Ours) | **66.27** | **73.77** | **0.77** | **0.89** | **0.43** |

ity questions to 50 unbiased users (age 20-50, randomized gender, AMT). The average rating was 5.8 (on a scale where 0 is "extremely unlikely" and 10 is "extremely likely"), indicating our proposed loss retains subject fidelity in most cases. We also had 50 unbiased users rank our method against baselines (i.e., "DiffNat", "DreamBooth", "Custom diffusion", "None is satisfactory"), totaling 1500 questionnaires. The aggregate responses showed that DiffNat-generated images significantly outperformed the baselines by a large margin (50.4%). Further details are provided in the Appendix.

### 4.2 TASK 2: UNCONDITIONAL IMAGE GENERATION

Unconditional image generation operates without the need for text or image guidance. It aims to learn the training data distribution through a generative model (in this case, a diffusion model) and produce samples that resemble the training data distribution. We opted for the well-known unconditional image generation pipeline, the DDPM Ho et al. (2020), to test the efficacy of KC loss. PG is especially effective here because classifier-free guidance cannot be applied.

In DDPM, we directly integrate the KC loss into the image space, demonstrating the flexibility of our proposed loss. We experimented with the Oxford-flowers Nilsback & Zisserman (2006), CelebA-faces Zhang et al. (2020), CelebAHQ Karras et al. (2017), Stanford-Dogs Khosla et al. (2011) and Stanford-Cars Krause et al. (2013) datasets, achieving consistent improvements in image quality, as shown in Table 2 and Figure 5. Additionally, PG further enhances image quality, as indicated in Table 2. Human evaluation is not feasible for unconditional image generation due to the lack of one-to-one correspondence between training and generated images, but quantitative and qualitative analyses demonstrate the effectiveness of our approach.

Table 2: Comparison of unconditional image generation task

| Method | Oxford flowers | | Celeb-faces | | CelebAHQ | | Stanford-Dogs | | Stanford-Cars | |
|---|---|---|---|---|---|---|---|---|---|---|
| | FID ↓ | MUSIQ ↑ | FID ↓ | MUSIQ ↑ | FID ↓ | MUSIQ ↑ | FID ↓ | MUSIQ ↑ | FID ↓ | MUSIQ ↑ |
| DDPM Ho et al. (2020) | 243.43 | 20.67 | 202.67 | 19.07 | 199.77 | 46.05 | 129.91 | 50.12 | 143.71 | 53.77 |
| DDPM Ho et al. (2020) + LPIPS | 242.62 | 20.80 | 201.55 | 19.21 | 197.17 | 46.15 | 115.72 | 50.86 | 137.22 | 53.98 |
| DDPM + KC loss (Ours) | 237.73 | 21.13 | 198.23 | 19.52 | 190.59 | 46.83 | 105.45 | 51.53 | 125.85 | 54.21 |
| DDPM + KC loss + PG (Ours) | **200.12** | **22.45** | **188.49** | **20.82** | **175.12** | **48.32** | **98.77** | **52.17** | **115.93** | **54.85** |

Table 4: Image super-resolution (x2) task

| Method | Image quality | | | | |
|---|---|---|---|---|---|
| | FID ↓ | PSNR ↑ | SSIM ↑ | LPIPS ↓ | MUSIQ ↑ |
| GD Dhariwal & Nichol (2021) | 100.2 | 19.4 | 0.62 | 0.25 | 58.12 |
| GD + KC loss (Ours) | 80.9 | 20.2 | 0.66 | 0.20 | 59.91 |
| GD + KC loss + PG (Ours) | 71.3 | 21.7 | **0.69** | **0.18** | **60.32** |
| LD. Karras et al. (2022) | 82.45 | 21.2 | 0.64 | 0.24 | 60.23 |
| LD + KC loss (Ours) | 70.12 | 22.3 | 0.70 | 0.18 | 62.15 |
| LD + KC loss + PG (Ours) | **59.32** | **23.6** | **0.73** | **0.16** | **63.72** |

Table 5: Image super-resolution (x8) task

| Method | Image quality | | | | |
|---|---|---|---|---|---|
| | FID ↓ | PSNR ↑ | SSIM ↑ | LPIPS ↓ | MUSIQ ↑ |
| GD Dhariwal & Nichol (2021) | 140.3 | 17.5 | 0.52 | 0.32 | 55.26 |
| GD + KC loss (Ours) | 125.5 | 18.7 | 0.56 | 0.27 | 57.33 |
| GD + KC loss + PG (Ours) | 108.6 | 19.1 | 0.58 | 0.23 | **58.62** |
| LD. Karras et al. (2022) | 103.2 | 18.7 | 0.59 | 0.25 | 58.62 |
| LD + KC loss (Ours) | 80.1 | 19.5 | 0.67 | 0.20 | 60.31 |
| LD + KC loss + PG (Ours) | **67.3** | **20.8** | **0.69** | **0.17** | **61.87** |

## 4.3 TASK 3: IMAGE SUPER-RESOLUTION

Image super-resolution typically takes the form of a conditional generation task, leveraging a low-resolution image as an additional condition for the diffusion model. In this study, we use two state-of-the-art diffusion pipelines as baselines for comparison. Guided diffusion (GD) Dhariwal & Nichol (2021) directly takes the low-

Table 3: Comparison of image super-resolution (x4) task

| Method | Image quality | | | | |
|---|---|---|---|---|---|
| | FID ↓ | PSNR ↑ | SSIM ↑ | LPIPS ↓ | MUSIQ ↑ |
| GD Dhariwal & Nichol (2021) | 121.23 | 18.13 | 0.54 | 0.28 | 57.31 |
| GD Dhariwal & Nichol (2021) + LPIPS | 119.81 | 18.22 | 0.54 | 0.27 | 57.42 |
| GD + KC loss (Ours) | 103.19 | 18.92 | 0.55 | 0.26 | 58.69 |
| GD + KC loss + PG (Ours) | **93.45** | **20.17** | **0.58** | **0.23** | **60.13** |
| LD. Karras et al. (2022) | 95.83 | 19.16 | 0.56 | 0.26 | 59.57 |
| LD. Karras et al. (2022) + LPIPS | 92.77 | 19.42 | 0.57 | 0.25 | 59.82 |
| LD + KC loss (Ours) | 83.34 | 20.25 | 0.58 | 0.22 | 61.20 |
| LD + KC loss + PG (Ours) | **71.33** | **21.92** | **0.60** | **0.19** | **62.85** |

resolution image as a condition and performs the diffusion operation in the pixel space. Additionally, we also explore the latent diffusion model (LD) Karras et al. (2022) that operates in the latent space of a pre-trained VQVAE Esser et al. (2021).

Note that, as GD operates in the pixel space, we directly add the proposed KC loss to the output of the denoising UNet. Conversely, for LD, we initially convert the latent embedding to image space using the pre-trained decoder and integrate the KC loss on the output of the decoder. For training, we use the standard FFHQ dataset Karras et al. (2017), which contains 70k high-quality images. We address the task of $\times 2$, $\times 4$, and $\times 8$ super-resolution where the GT images are of resolution $256 \times 256$. We evaluate randomly sampled 3000 images from CelebA-Test dataset Karras et al. (2017) under the same $\times 2$, $\times 4$ and $\times 8$-SR setting in Tab. 4, Tab. 3 and Tab. 5 respectively. In the qualitative results shown in Fig. 6, we observe that adding KC loss improves the image quality and finer details, e.g., eye structure, texture, and lighting.

**Human evaluation.** We conduct a human evaluation of the image super-resolution task to compare GD and LD with the addition of KC loss to each counterpart (DiffNat). The aggregate response of choices (corresponding to best quality images w.r.t methods) from 50 unbiased users (age 20-50, randomized gender, AMT) across 1000 questionnaires, shown in Fig. 7, indicates that DiffNat-generated images have superior quality compared to the GD and LD baselines.

**Other tasks.** Additionally, we verify efficacy of KC loss for blind face restoration Suin et al. (2024) (Sec. L.1) and one-shot video editing Wu et al. (2023) task (Sec. L.2) as shown in the Appendix.

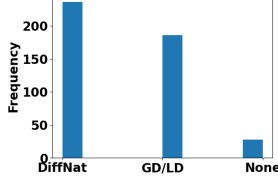

Figure 7: Human evaluation for image super-resolution task.

Table 6: Loss & guidance ablation on DB (SD-1.5)

| KC | CFG | PG | FID ↓ | MUSIQ ↑ |
|---|---|---|---|---|
| ✗ | ✗ | ✗ | 125.56 | 67.12 |
| ✗ | ✗ | ✓ | 114.38 | 68.05 |
| ✗ | ✓ | ✗ | 111.76 | 68.31 |
| ✗ | ✓ | ✓ | 109.12 | 68.92 |
| ✓ | ✗ | ✗ | 105.33 | 69.34 |
| ✓ | ✓ | ✗ | 100.08 | 69.78 |
| ✓ | ✗ | ✓ | 98.21 | 70.02 |
| ✓ | ✓ | ✓ | **93.45** | **70.82** |

Table 7: Loss & guidance ablation on DB (SDXL)

| KC | CFG | PG | FID ↓ | MUSIQ ↑ |
|---|---|---|---|---|
| ✗ | ✗ | ✗ | 102.32 | 70.18 |
| ✗ | ✗ | ✓ | 96.17 | 70.92 |
| ✗ | ✓ | ✗ | 95.32 | 71.35 |
| ✗ | ✓ | ✓ | 89.72 | 71.87 |
| ✓ | ✗ | ✗ | 85.11 | 72.05 |
| ✓ | ✓ | ✗ | 80.75 | 72.32 |
| ✓ | ✗ | ✓ | 73.33 | 72.88 |
| ✓ | ✓ | ✓ | **70.18** | **73.02** |

## 5 ABLATION AND ANALYSIS

**Ablation of loss and guidance.** Here we perform ablation studies of diffusion backbone (SD-1.5, SDXL), KC loss and PG as shown in Tab. 6 & 7 on DreamBooth dataset. We observe that PG is complementary to classifier-free guidance (CFG) and both KC loss and PG improve image quality as shown in Fig. 4, Fig. 5, and Fig. 6. We have also performed ablations w.r.t transforms (DCT, DWT)

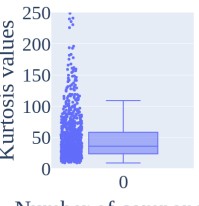 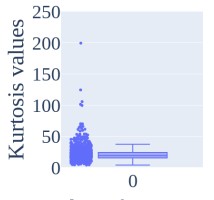 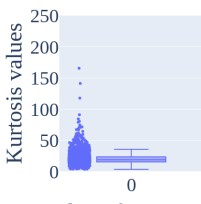

(a) Avg. kurtosis of GD generated images

(b) Avg. kurtosis of images trained with GD + KC loss

(c) Avg. kurtosis of Natural images

Figure 8: Average kurtosis analysis of guided diffusion (GD) framework trained on FFHQ dataset. From this analysis, it is evident that GD-generated images have higher kurtosis deviation. Integrating KC loss reduces the kurtosis deviation to preserve the naturalness of the generated images. Natural images have more concentrated kurtosis values.

Table 9: PAR analysis of tasks - DB, CD, DDPM on Oxford flowers (OF), DDPM on CelebFaces (CF), DDPM on CelebAHQ (CelebHQ), GD on FFHQ, LD on FFHQ has been reported.

| Setting | DB | CD | DDPM (OF) | DDPM (CF) | DDPM (CelebHQ) | GD | LD |
|---|---|---|---|---|---|---|---|
| w/o KC loss | 1.64 | 0.63 | 3.02 | 7.09 | 3.20 | 0.89 | 1.11 |
| w KC loss | **0.75** | **0.36** | **2.99** | **6.97** | **2.63** | **0.51** | **1.07** |

and the results are shown in Tab. 11 (Appendix).

**Kurtosis analysis.** To verify the efficacy of the proposed KC loss, we performed an average kurtosis analysis by computing the average kurtosis deviation of DWT-filtered images from the FFHQ dataset and plotting the results in Fig. 8. The analysis showed that images generated with GD had the highest kurtosis deviation (Fig. 8 (a)), while natural images had the least deviation (Fig. 8 (c)), and adding KC loss reduced the kurtosis deviation (Fig. 8 (b)), thus improving image quality as demonstrated both qualitatively and quantitatively.

**Comparison of real vs synthetic detection.** To analyze the robustness of the proposed KC loss, we train a classifier to distinguish real images from synthetic ones generated by diffusion models, including those with and without KC loss. The results in Tab. 8 show that adding KC loss decreased the real vs synthetic classification accuracy, indicating that the generated images with KC loss have higher perceptual quality and appear more natural to both human viewers and machine algorithms.

Table 8: Comparison of real vs synthetic detection

| Method | Accuracy |
|---|---|
| DB | 93.33% |
| DB + KC loss | 66.66% |
| CD | 94.16 % |
| CD + KC loss | 92.5% |

**Perceptual artifact analysis.** Zhang et al. (2023) identified perceptual artifacts in diffusion-generated images, which adversely impact image quality, and developed a dataset and metric (Perceptual Artifacts Ratio, PAR) to automate artifact localization/editing. Our analysis shows that incorporating KC loss reduces these perceptual artifacts (Fig. 9), as evidenced by a decrease in average PAR (Table. 9), demonstrating that KC loss inherently enhances image quality by minimizing artifacts.

**Limitations.** The proposed PG strategy necessitates two forward passes through the diffusion model to obtain the guidance direction, which is time-consuming. We aim to address this issue in future work. Some failure cases are also presented in the Appendix.

## 6 CONCLUSION

While diffusion models have made significant strides in generating naturalistic images, enhancing image quality remains a key focus. We introduce a novel and generic KC loss, leveraging the KC property of natural images, which minimizes the gap between maximum and minimum kurtosis values across different DWT-filtered versions of the image. Additionally, we

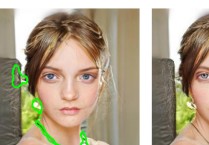 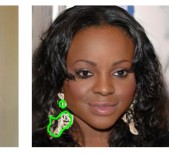 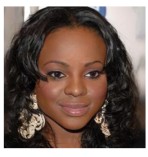

(a) GD  (b) GD + KC  (c) LD  (d) LD + KC

Figure 9: Perceptual artifact ratio analysis. Green boundaries localizes perceptual artifacts. Adding KC loss reduces such artifacts

propose a condition-agnostic PG strategy to further improve image quality. Our experiments show that KC loss and PG improve image quality in various generative tasks, including personalized few-shot fine-tuning of text-to-image models, unconditional image generation, and image super-resolution. Human studies validate the effectiveness of our approach.

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

## A APPENDIX

In this supplementary material, we will provide the following details.

1. Training details.
2. Theoretical justification.
3. KC loss added as a regularizer.
4. Additional Ablations.
5. Failure cases.
6. Kurtosis analysis.
7. Computational complexity.
8. Convergence analysis.
9. Qualitative analyis.
10. Experiments on image super-resolution. Experiments on other tasks.

## B TRAINING DETAILS

The training details of finetuning the diffusion model for various tasks have been provided here. For personalized few-shot finetuning, we consider two methods - Dreambooth Ruiz et al. (2022) and Custom diffusion Kumari et al. (2022). For fair comparison, we applied both the approaches on the dataset and setting introduced by Dreambooth. The dataset contains 30 subjects (e.g., backpack, stuffed animal, dogs, cats, sunglasses, cartoons etc) and 25 prompts including 20 re-contextualization prompts and 5 property modification prompts. DINO, which is the average pairwise cosine similarity between the ViT-S/16 DINO embeddings Caron et al. (2021) of the generated and real images. (2) CLIP-I, i.e., the average pairwise cosine similarity between CLIP Radford et al. (2015) embeddings of the generated and real images. To measure the prompt fidelity, we use CLIP-T, which is the average cosine similarity between prompt and image CLIP embeddings.

For unconditional image generation, we have experimented on oxford flowers, CelebAfaces and CelebAHQ datasets. Image quality has been measured by FID and MUSIQ score.

In case of image super-resolution, we experimented with guided diffusion Dhariwal & Nichol (2021) and latent diffusion Karras et al. (2022) pipelines. We use FFHQ dataset for training, and test on a subset of 1000 images from CelebAHQ test set for x4 super-resolution task. The hyperparameter details are given in Tab. 10.

Table 10: Hyperparameters

| Hyperparameter | Values |
| --- | --- |
| Coefficient of $L_{recon}$ | 1 |
| Coefficient of $L_{prior}$ | 1 |
| Coefficient of $L_{KC}$ | 1 |
| Perceptual guidance scale | 1.001 |
| Learning rate | $10^{-5}$ |
| Batch size (Dreambooth, Custom diffusion) | 8 |
| Batch size (DDPM) | 125 |
| Batch size (GD) | 16 |
| Batch size (LD) | 9 |
| Text-to-image diffusion model | Stable Diffusion-v1 Rombach et al. (2022) |
| Number of class prior images (Dreambooth, Custom diffusion ) | 10 |
| Number of DWT components | 25 |
| DWT filter | Daubechies |

## C THEORETICAL JUSTIFICATION

Here we provide theoretical analysis of the Lemmas mentioned in the main paper.

**Definition 2** *Lipschitz Continuity : A function $f$ is said to be Lipschitz continuous if there exists a constant L (called the Lipschitz constant) such that for all $x$ and $y$ in the domain of $f$:*

$$|f(x) - f(y)| \leq L|x - y| \tag{9}$$

**Definition 3** *Max-Min Difference : Consider the function $f$ :*

$$f(\kappa_1, \kappa_2, \ldots, \kappa_n) = \max(\kappa_i) - \min(\kappa_i) \tag{10}$$

**Definition 4** *Lipschitz Condition: We need to show that there exists a constant L such that for any two sets of kurtosis values $(\kappa_1, \kappa_2, \ldots, \kappa_n)$ and $(\kappa'_1, \kappa'_2, \ldots, \kappa'_n)$,*

$$|f(\kappa_1, \kappa_2, \ldots, \kappa_n) - f(\kappa'_1, \kappa'_2, \ldots, \kappa'_n)| \leq L \sum_{i=1}^{n} |\kappa_i - \kappa'_i| \tag{11}$$

**Lemma 4** *KC loss is differentiable and Lipschitz continuous with Lipschitz constant 2.*

*Proof.* We are taking maximum and minimum across kurtosis values, therefore KC loss is differentiable and so as the combined loss, since differentiability preserves over addition.

Next, we proof the Lipschitz continuity. Note, the function $\max(\kappa_i)$ is 1-Lipschitz because:

$$|\max(\kappa_i) - \max(\kappa'_i)| \leq \max_i |\kappa_i - \kappa'_i| \tag{12}$$

Similarly, the function $\min(\kappa_i)$ is also 1-Lipschitz because:

$$|\min(\kappa_i) - \min(\kappa'_i)| \leq \max_i |\kappa_i - \kappa'_i| \tag{13}$$

Since both the maximum and minimum functions are 1-Lipschitz, their difference is also Lipschitz continuous with a constant of 2:

$$|f(\kappa_1, \kappa_2, \ldots, \kappa_n) - f(\kappa'_1, \kappa'_2, \ldots, \kappa'_n)| \leq 2 \max_i |\kappa_i - \kappa'_i| \tag{14}$$

For simplicity, if we consider the $l_1$ norm of the differences, we get:

$$|f(\kappa_1, \kappa_2, \ldots, \kappa_n) - f(\kappa'_1, \kappa'_2, \ldots, \kappa'_n)| \leq 2 \sum_{i=1}^{n} |\kappa_i - \kappa'_i| \tag{15}$$

Thus, we have shown that the KC loss function, i.e., the difference between the maximum and minimum kurtosis values of wavelet-transformed coefficients of natural images is Lipschitz continuous with a Lipschitz constant of 2 when considering the $l_1$ norm.

**Lemma 5** *A Gaussian scale mixture (GSM) vector $x$ with zero mean has the following probability density function:*

$$p(x) = \int_0^\infty \mathcal{N}(x; 0, z\Sigma_x) p_z(z) dz \tag{16}$$

*and its projection kurtosis is __constant__ with respect to the projection direction w, i.e.,*

$$\kappa(w^T x) = \frac{3 var_z\{z\}}{\mathcal{E}_z\{z\}^2} \tag{17}$$

*where $\mathcal{E}_z\{z\}$ and $var_z\{z\}$ are the mean and variance of latent variable $z$ respectively.*

*Proof.* Marginal distribution of the projection of $x$ on non-zero vector $w$ is given by Zhang & Lyu (2014),

$$p_w(t) = \int_{x:w^T x = t} p(x) dx$$

$$= \int_z p_z(z) dz . \int_{x:w^T x = t} \frac{1}{\sqrt{(2\pi z)^d |det(\Sigma_x)|}} exp(-\frac{x^T \Sigma_x^{-1} x}{2z}) dx$$

$$= \int_z \mathcal{N}_t(0, z w^T \Sigma_x w) p_z(z) dz$$

Note that, the last equality holds from the marginalization property of Gaussian, i.e., $X \approx \mathcal{N}(\mu, \Sigma)$, then, $AX \approx \mathcal{N}(A\mu, A\Sigma A^T)$.

The variance of $w^T x$,

$$
\begin{aligned}
\mathcal{E}_t\{t^2\} &= \int_z p_z dz \int_t t^2 \mathcal{N}_t(0, zw^T\Sigma_x w)dz \\
&= w^T\Sigma_x w \int_z z p_z dz \\
&= w^T\Sigma_x w \mathcal{E}_z\{z\}
\end{aligned}
$$

The fourth order moment of $w^T x$,

$$
\begin{aligned}
\mathcal{E}_t\{t^4\} &= \int_z p_z dz \int_t t^4 \mathcal{N}_t(0, zw^T\Sigma_x w)dz \\
&= 3(w^T\Sigma_x w)^2 \int_z z^2 p_z dz \\
&= 3(w^T\Sigma_x w)^2 \mathcal{E}_z\{z^2\}
\end{aligned}
$$

We utilize the property that $\mathcal{N}_t(0, \sigma^2)$ has a fourth order moment of $3\sigma^4$.

Finally, the kurtosis becomes,

$$
\begin{aligned}
\kappa(w^T x) &= \frac{\mathcal{E}_t\{t\}^4}{\mathcal{E}_t\{t\}^2} - 3 \\
&= \frac{3\mathcal{E}_z\{z\}^2}{\mathcal{E}_z\{z\}^2} - 3 \\
&= \frac{3(\mathcal{E}_z\{z^2\} - \mathcal{E}_z\{z\}^2)}{\mathcal{E}_z\{z\}^2} \\
&= \frac{3var_z\{z\}}{\mathcal{E}_z\{z\}^2}
\end{aligned}
$$

**Lemma 6** *If the noisy version of the natural image is denoted by, y = x + n, where x is a whitened GSM vector (normalized natural image) and n is a zero-mean white Gaussian noise with variance $\sigma^2 I$, x and n are mutually independent of each other, then the projection kurtosis of y, $\kappa(w^T y)$ can be expressed as:*

$$
\kappa(w^T y) = \kappa(w^T x)\Big(1 - \frac{c}{SNR(y)}\Big)^2 = \frac{3var_z\{z\}}{\mathcal{E}_z\{z\}^2}\Big(1 - \frac{c}{SNR(y)}\Big)^2 \tag{18}
$$

*where Signal-to-Noise Ratio (SNR) is defined as, $SNR(y) = \frac{\sigma^2(y)}{\sigma^2(n)}$ and c is a constant.*

*Proof.* Here, we provide the proof of Lemma 1, mentioned in the main paper. Without loss of generality, we strat by assuming, $\mathcal{E}_x x = 0$, since the mean can be easily subtracted from the data. We also assume that n is a zero-mean white Gaussian noise with variance $\sigma^2 I$, x and n are mutually independent of each other.

$$
\begin{aligned}
\sigma^2(w^T n) &= w^T \mathcal{E}_z\{zz^T\}w = \sigma^2 w^T w = \sigma^2 \\
\sigma^2(w^T x) &= w^T \mathcal{E}_x\{xx^T\}w = w^T\Sigma_x w \\
\sigma^2(w^T y) &= \sigma^2(w^x y) + \sigma^2(w^T n) = w^T\Sigma_x w + \sigma^2
\end{aligned}
$$

Since $n$ is a white Gaussian, $x$ and $n$ are independent, then $w^T x$ and $w^T n$ Therefore,

$$
\sigma^2(w^T y) = \sigma^2(w^T x) + \sigma^2(w^T n) \tag{19}
$$

.

Similarly, for fourth order moment, using the additivity of cumulants of independent variables (since $x$ and $n$ are independent) Papoulis & Unnikrishna Pillai (2002), we obtain,

$$\kappa(w^T y)(\sigma^2(w^T y))^2 = \kappa(w^T x)(\sigma^2(w^T x))^2 + \kappa(w^T n)(\sigma^2(w^T n))^2$$
$$= \kappa(w^T x)(\sigma^2(w^T x))^2 \tag{20}$$

Since, For Gaussian, $\kappa(n) = 0$

By rearranging, we have,

$$\kappa(w^T y) = \kappa(w^T x).\left(\frac{\sigma^2(w^T x)}{\sigma^2(w^T y)}\right)^2$$

$$= \kappa(w^T x).\left(\frac{\sigma^2(w^T y) - \sigma^2}{\sigma^2(w^T y)}\right)^2$$

$$= \kappa(w^T x).\left(1 - c.\frac{\sigma^2}{\sigma^2(y)}\right)^2$$

$$= \frac{3 var_z\{z\}}{\mathcal{E}_z\{z\}^2}.\left(1 - \frac{c}{SNR(y)}\right)^2$$

Here, Signal-to-Noise Ratio (SNR) is defined as, $SNR(y) = \frac{\sigma^2(y)}{\sigma^2(n)}$.

## D    KC LOSS ADDED AS A REGULARIZER

We would like to highlight that in our work, the underlying theoretical framework behind the forward and reverse diffusion processes remains unchanged; rather, we focus on improving the performance of the denoising neural network used to approximate the reverse diffusion trajectory.

Suppose, we have the input training images ($x$) and conditioning vector $c$. The conditioning vector could be text (text-to-image model), image (image-to-image model), or none (in case of the unconditional diffusion model). In the forward process, the noisy versions of image $x$ at timestep $t$ is generated as $x_t = \alpha_t x + \sigma_t \epsilon$, where $\epsilon \sim N(0, I)$.

In the reverse process, a denoised autoencoder ($f_\theta$) is trained to predict the denoised version of the image ($x_{t,gen}$) at each timestep $t$ from the noisy images $x_t$, i.e., $x_{t,gen} = f_\theta(x_t, c, t)$. Typically, the denoised autoencoder ($f_\theta$) is trained by minimzing the Mean Squared Error between the real image ($x$) and the generated denoised version of the image at time step $t$ ($x_{t,gen}$) averaged over timesteps and noise variances as denoted by,

$$L_{recon} = \mathbb{E}_{x,c,\epsilon,t}[\,||x_{t,gen} - x||_2^2] \tag{21}$$

The kurtosis concentration loss is applied on the generated images ($x_{t,gen}$), and therefore can be considered as a function ($f'$) of $x_{gen}$ as follows:

$$L_{KC} = \mathbb{E}_{x,c,\epsilon,t}[f'(x_{t,gen})] \tag{22}$$

Note the function $f'$ is difference between the maximum and minimum values of the DWT filtered version of input $x_{t,gen}$.

Therefore, the total loss function can be written as,

$$L_{total} = \mathbb{E}_{x,c,\epsilon,t}[\,||x_{t,gen} - x||_2^2] + \mathbb{E}_{x,c,\epsilon,t}[f'(x_{t,gen})] \tag{23}$$

In our work, the above-mentioned framework remains the same. Instead, the proposed KC loss acts as an additional regularizer to the training of the denoising neural network, which helps it to denoise $x_t$ better (Lemma 2, main paper), ultimately improving the approximation of $x$, i.e., $x_{t,gen}$ at each time step t.

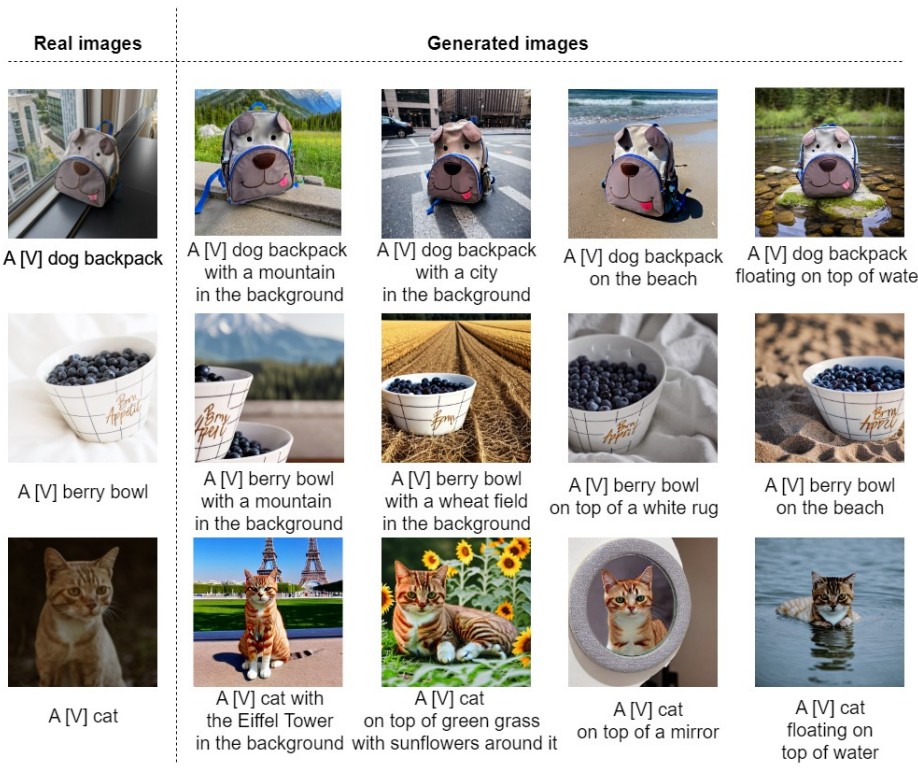

Figure 10: DiffNat generated images. The task is to learn a unique identifier ("A [V] dog backpack") of the training images and generate variations w.r.t. background, lighting conditions etc. The generated images look natural in different background context, e.g., "A [V] dog backpack on the beach/ with a city in the background etc". The generated images are of high quality.

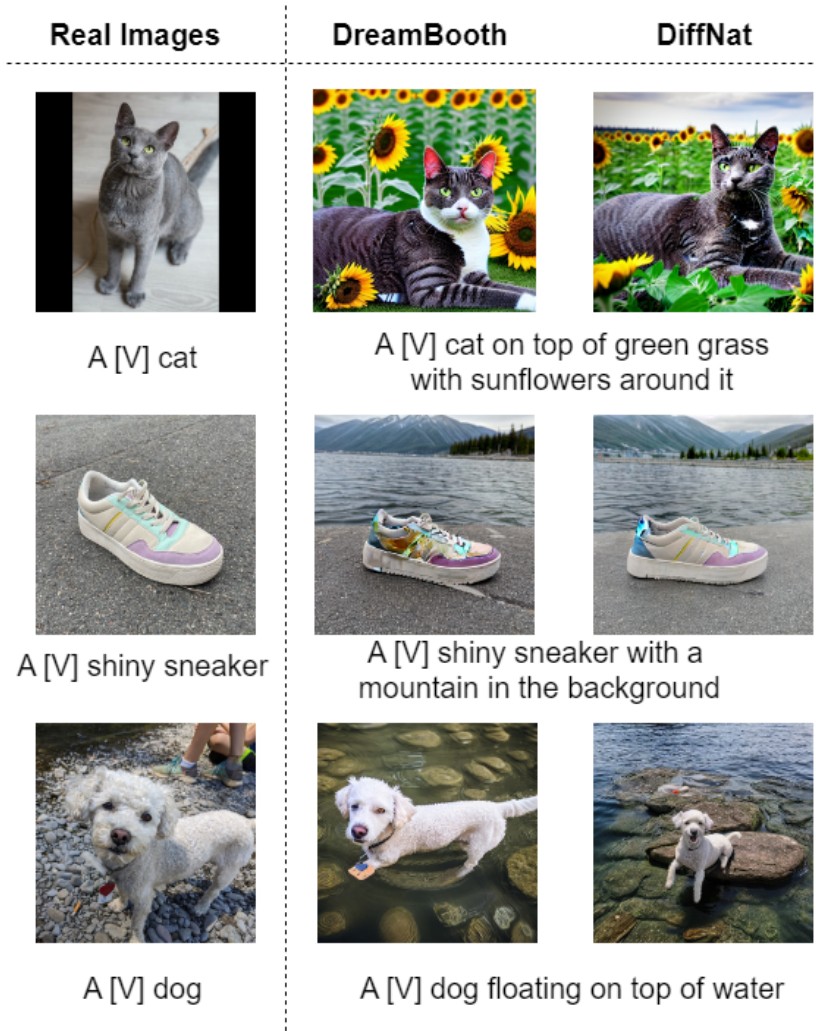

Figure 11: Comparison of DreamBooth and DiffNat. DiffNat generated images have better visual quality.

## E    ADDITIONAL ABLATIONS

In Fig. 10, we visualize some of the DiffNat generated images using various text-prompts. The generated images capture the context of the text-prompt and also retain naturalness. We have also provided qualitative comparison w.r.t Dreambooth in Fig. 11.

We also provide ablations for using DCT transforms and analyse the performance with respect to other tasks and methods. Experiments in Table. 11 suggests DWT performs better than DCT for different methods across datasets.

## F    FAILURE CASES

We also present some of the failure cases of DiffNat in Fig. 12. E.g., our model fails to generate images of "A [V] berry bowl with the Eiffel Tower in the background", but actually generates images with "the Eiffel Tower" in the berry bowl. Similarly, the model fails to generate "A cube shaped [V] can", since these object do not appear in the training set. The model also fails to generate "A [V] cat on top of a purple rug in a forest" and instead generated some version of purple cat.

Figure 12: Failure cases of DiffNat. Instead of generating "A [V] berry bowl with the Eiffel Tower in the background", our method generates image with the Eiffel Tower in the berry bowl. Also, while generating "A [V] cat on top of a purple rug in a forest", it generates a purple [V] cat, which shows the color bias w.r.t the text-prompt of the model.

Table 11: Comparison of DCT vs DWT

| Method | FID score ↓ | MUSIQ score ↑ |
|---|---|---|
| DB (Dreambooth dataset) | 111.76 | 68.31 |
| DB + KC (DCT) | 106.23 | 68.72 |
| DB + KC (DWT) | **100.08** | **69.78** |
| CD (Dreambooth dataset) | 84.65 | 70.15 |
| CD + KC (DCT) | 80.33 | 70.67 |
| CD + KC (DWT) | **75.68** | **72.22** |
| DDPM (Oxford flowers) | 243.43 | 20.67 |
| DDPM + KC (DCT) | 240.12 | 20.98 |
| DDPM + KC (DWT) | **237.73** | **21.13** |
| GD (FFHQ) | 121.23 | 57.31 |
| GD + KC (DCT) | 112.66 | 58.12 |
| GD + KC (DWT) | **103.19** | **58.69** |
| LD (FFHQ) | 95.83 | 59.57 |
| LD + KC (DCT) | 88.52 | 60.37 |
| LD + KC (DWT) | **83.34** | **61.20** |

## G  KURTOSIS ANALYSIS

To verify the efficacy of the proposed KC loss, we perform average kurtosis analysis in this section. we compute the average kurtosis deviation of DWT filtered version of images from the dataset and plot them in Fig. 14, Fig. 15 and Fig. 16. E.g., in case of dreambooth task, we compute the kurtosis statistics of bandpass filtered version of natural images from Dreambooth dataset, images generated by Dreambooth and images generated by DiffNat (i.e., adding KC loss) and plot it in Fig. 14. We observe that the Dreambooth generated images (Fig. 14 (a)) have highest kurtosis deviation. The average deviation is least for natural images (Fig. 14 (c)) and adding KC loss reduces the kurtosis deviation (Fig. 14 (b)). Similar trends can be observed for DDPM (Fig. 15), guided diffusion (Fig. 16) as well. Adding KC loss improves image quality has been verified both qualitatively and quantitatively in the paper. This analysis verifies minimizing kurtosis loss improves diffusion image quality.

## H  COMPUTATIONAL COMPLEXITY

Here we analyze the computational complexity of the proposed KC loss. Suppose, given a batch of N images. We need to perform DWT of each images using k different filters. Since, DWT for 'Haar'/'Daubechis' wavelet can be done in linear time, the complexity of performing DWT with k filters can be done in $\mathcal{O}(Nk)$ time. Now, calculating the difference between maximum and minimum kurtosis can be done in linear time, therefore, the computational complexity of calculating KC loss is $\mathcal{O}(Nk)$. This minimal overhead of computing KC loss can be observed in the training time analysis provided next. The run time analysis has been provided in Table. 12. Note that the experiments for Dreambooth, Custom diffusion, DDPM have been performed on a single A5000 machine with 24GB GPU. We have performed guided diffusion (GD) and latent diffusion (LD) experiments on a server of 8 24GB A5000 GPUs. The experimental results in Table. 12 show that incorporating KC loss induces minimum training overhead.

Table 12: Training time analysis

| Method | dataset | Training time |
|---|---|---|
| DreamBooth Ruiz et al. (2022) | 5-shot finetuning | 10 min 21s |
| DreamBooth Ruiz et al. (2022) + KC loss | 5-shot finetuning | 11 min 30s |
| Custom Diffusion Kumari et al. (2022) | 5-shot finetuning | 6m 43s |
| Custom Diffusion Kumari et al. (2022) + KC loss | 5-shot finetuning | 7m 11s |
| DDPM Ho et al. (2020) | CelebAfaces | 2d 8h 21m |
| DDPM Ho et al. (2020) + KC loss | CelebAfaces | 2d 9h 19m |
| GD Dhariwal & Nichol (2021) | FFHQ | 23h 10m |
| GD Dhariwal & Nichol (2021) + KC loss | FFHQ | 1d 1h 29m |
| LD Karras et al. (2022) | FFHQ | 20h 15m |
| LD Karras et al. (2022) + KC loss | FFHQ | 22h 40m |

## I  CONVERGENCE ANALYSIS

The main idea of the diffusion model is to train a UNet, which learns to denoise from a random noise to a specific image distribution. More denoising steps ensure a better denoised version of the image, e.g., DDPM Ho et al. (2020), LDM Karras et al. (2022). In proposition 1 (main paper), we show that minimizing projection kurtosis further denoise input signals. Therefore, KC loss helps in the denoising process and improves the convergence speed. We have shown that adding KC loss improves the loss to converge faster for Dreambooth task in Fig. 13.

## J  QUALITATIVE ANALYSIS

In this section, we provide more qualitative analysis to show that adding KC loss improves image quality. Zoomed view of the generated images are shown to compare w.r.t the baselines in Fig. 18, Fig. 19, Fig. 20, Fig. 21, Fig. 22, Fig. 23, Fig. 24, Fig. 25. Details are provided in the caption.

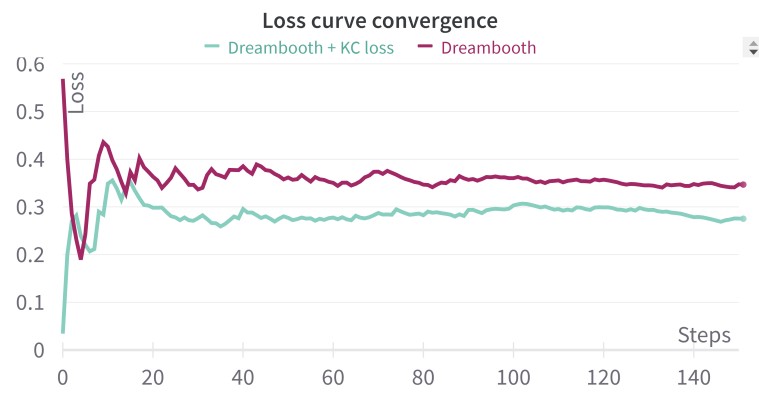

Figure 13: Loss curve convergence of Dreambooth.

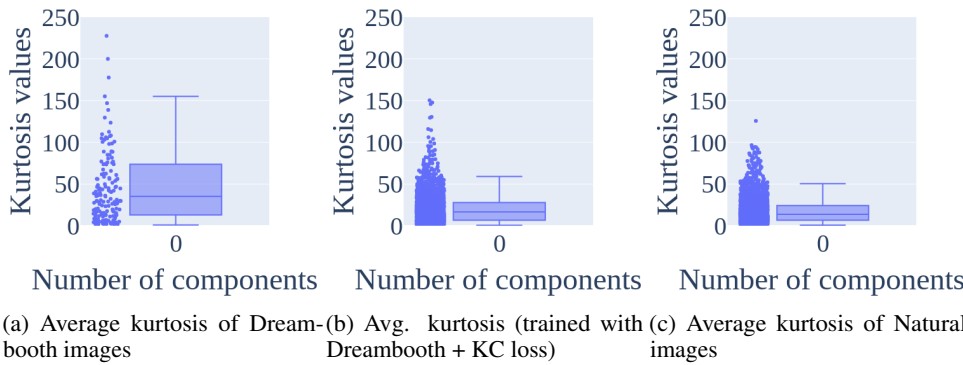

(a) Average kurtosis of Dream- (b) Avg. kurtosis (trained with (c) Average kurtosis of Natural
booth images                    Dreambooth + KC loss)            images

Figure 14: Average kurtosis analysis of Dreambooth, DiffNat and natural images over the dataset used in Dreambooth. From this analysis, it is evident that Dreambooth generated images have higher kurtosis deviation. Integrating KC loss reduces the kurtosis deviation to preserve the naturalness of the generated images. Natural images have more concentrated kurtosis values.

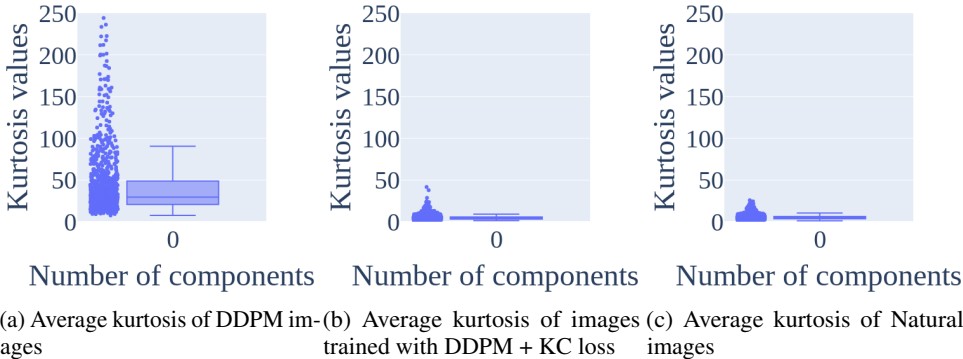

(a) Average kurtosis of DDPM im- (b) Average kurtosis of images (c) Average kurtosis of Natural
ages                             trained with DDPM + KC loss    images

Figure 15: Average kurtosis analysis of DDPM framework trained on Oxford flowers dataset. From this analysis, it is evident that DDPM generated images have higher kurtosis deviation. Integrating KC loss reduces the kurtosis deviation to preserve the naturalness of the generated images. Natural images have more concentrated kurtosis values.

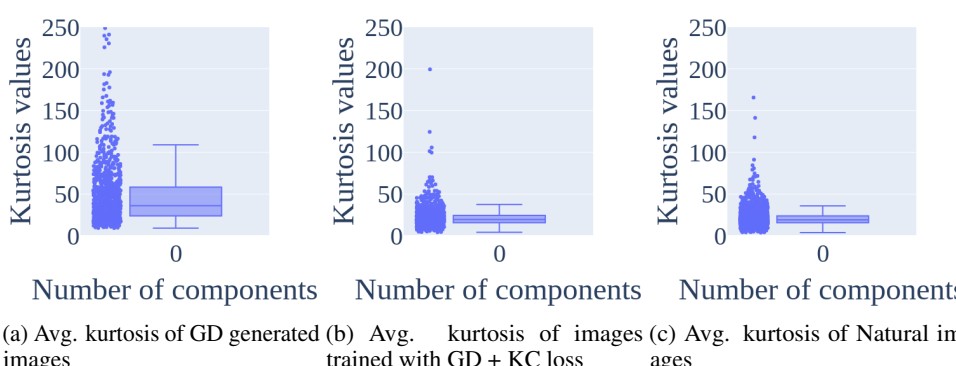

(a) Avg. kurtosis of GD generated images

(b) Avg. kurtosis of images trained with GD + KC loss

(c) Avg. kurtosis of Natural images

Figure 16: Average kurtosis analysis of guided diffusion (GD) framework trained on FFHQ dataset. From this analysis, it is evident that GD generated images have higher kurtosis deviation. Integrating KC loss reduces the kurtosis deviation to preserve the naturalness of the generated images. Natural images have more concentrated kurtosis values.

## K EXPERIMENTS ON IMAGE SUPER-RESOLUTION

In this section, we provide more experimental results for image super-resolution task. This includes quantitative results and human evaluation.

### K.1 QUANTITATIVE RESULTS

In addition to the super resolution task (x4) shown in the main paper, we conduct experiments for x2 and x8 tasks as well in the same setting. The ground-truth images are of size 256 X 256. Therefore, x2 task performs image super-resolution from 128 X 128 → 256 X 256 and x8 task performs image super-resolution from 32 X 32 → 256 X 256 and the corresponding experiments are shown in Table 13 and Table 14 respectively. For training, we use standard FFHQ dataset Karras et al. (2017), and evaluation is performed on CelebA-Test dataset Karras et al. (2017). We observe that adding KC loss improves image quality quantitatively both for guided diffusion (GD) and latent diffusion (LD). Qualitative results are shown in Fig. 22, Fig. 23, Fig. 24 and Fig. 25. Next, we also perform human study to validate our approach.

### K.2 HUMAN EVALUATION

We conduct human evaluation of image super-resolution task to compare guided diffusion (GD)/ latent diffusion (LD) and adding KC loss to the corresponding counterpart (DiffNat). We provide 20 examples of natural images and corresponding generated images using GD, LD and our method DiffNat (i.e., adding KC loss) and asked the following question to amazon mechanical turks: "which of the generated images is of best visual quality considering factors include image quality and preserving the identity of the original image?" Similar to Dreambooth task, we evaluate this by 50 users, totalling 1000 questionnaires. The available options are { 'DiffNat', 'GD/LD', 'None is satisfactory' }. The aggregate response shows that DiffNat generated images are of better image quality compared to the baselines, as shown in Fig. 26. Therefore, we verified the improved image quality quantitatively, qualitatively and through human evaluation as well. Note that, human evaluation is not applicable for unconditional image generation task since there is no one-to-one correspondence between the training images and the generated images. It will be ambiguous for the human observers to compare quality between approaches. Therefore, we abstain ourselves from performing human evaluation for this task. However, the quantitative and qualitative analysis exhibit the efficacy of our approach.

## L EXPERIMENTS ON OTHER TASKS

We analyze the effectiveness of KC loss on other visual recognition tasks, e.g., one-shot video editing and generation, and inverse problem like blind face restoration.

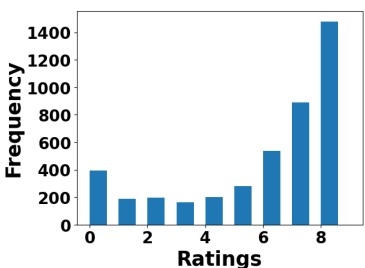

Figure 17: Subject fidelity assessment by user study. The ratings ranges from "0" being "extremely unlikely" to 10 being "extremely likely". We observe from the plot that most of the users find DiffNat preserves subject fidelity. The average rating is 5.8, which is "moderately likely" to "highly likely".

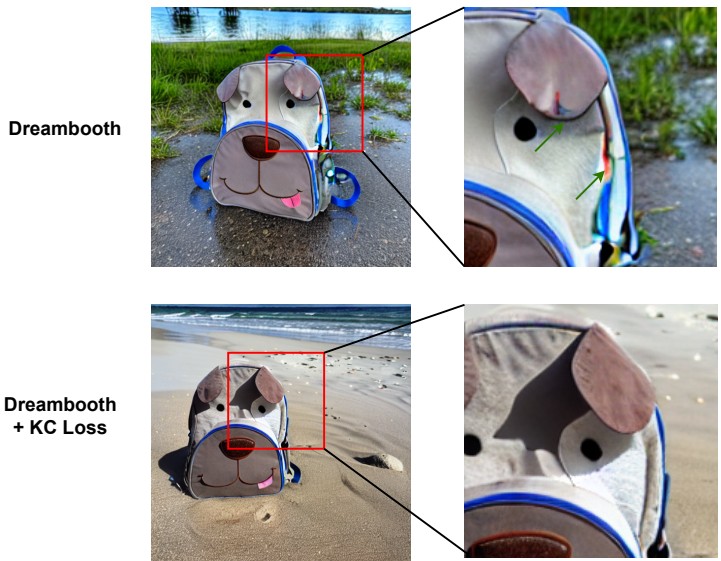

Figure 18: Qualitative comparison of with/without KC loss in Dreambooth. The bottom image (with KC loss) shows better image quality and shadows (best viewed in color).

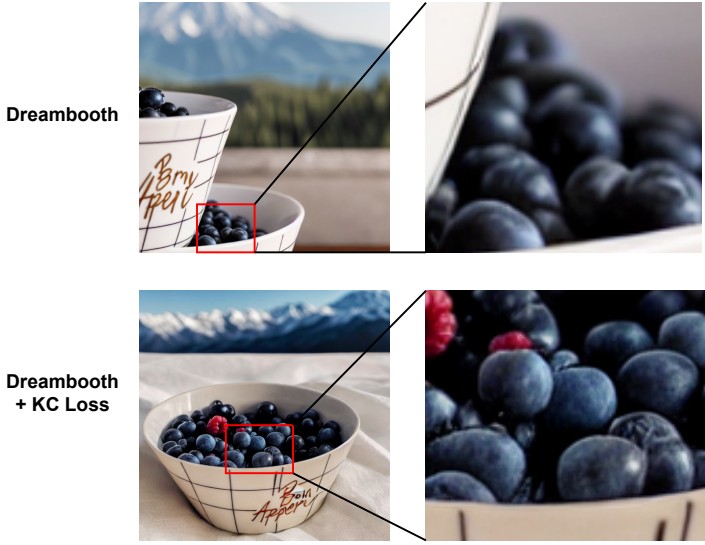

Figure 19: Qualitative comparison of with/without KC loss in Dreambooth. The bottom image (with KC loss) shows better image quality and reflections on the bowl full of berries (best viewed in color).

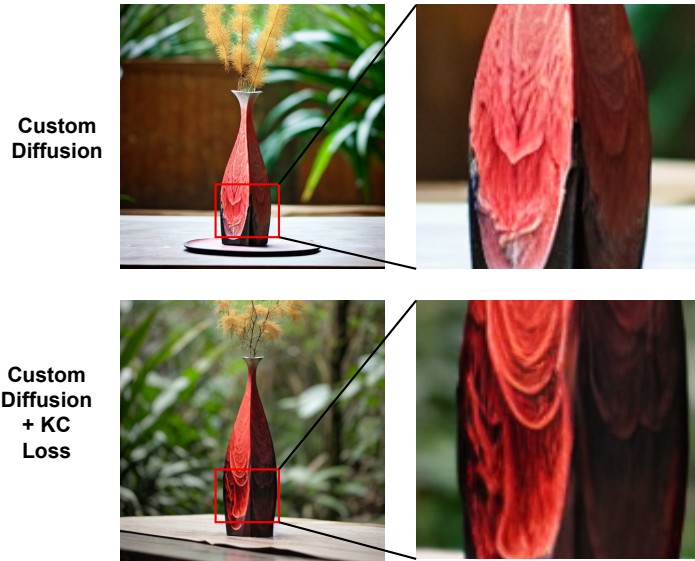

Figure 20: Qualitative comparison of with/without KC loss in Custom diffusion. The bottom image (with KC loss) shows better image quality in terms of color vividness and contrast (best viewed in color).

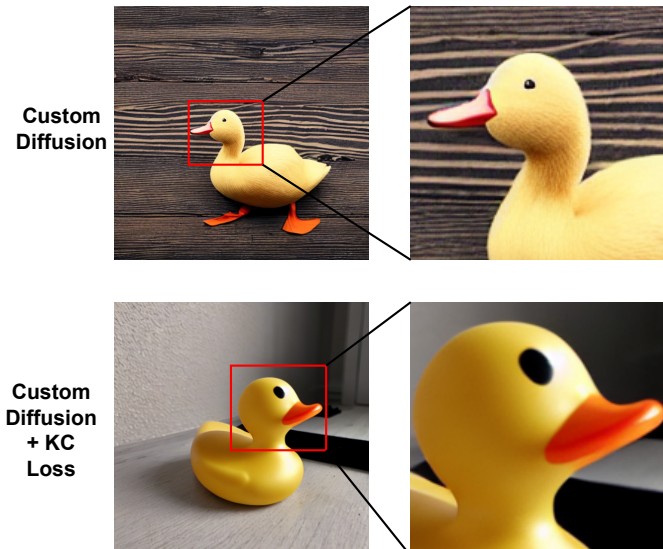

Figure 21: Qualitative comparison of with/without KC loss in Custom diffusion. The bottom image (with KC loss) shows better image quality in terms of detail and smoothness (best viewed in color).

Table 13: Comparison of image super-resolution (x2) task

| Method | Image quality | | | | |
|---|---|---|---|---|---|
| | FID score ↓ | PSNR ↑ | SSIM ↑ | LPIPS ↓ | MUSIQ score ↑ |
| GD Dhariwal & Nichol (2021) | 100.2 | 19.4 | 0.62 | 0.25 | 58.12 |
| GD + KC loss(Ours) | **80.9** | **20.2** | **0.66** | **0.20** | **59.91** |
| LD. Karras et al. (2022) | 82.45 | 21.2 | 0.64 | 0.24 | 60.23 |
| LD + KC loss(Ours) | **70.12** | **22.3** | **0.70** | **0.18** | **62.15** |

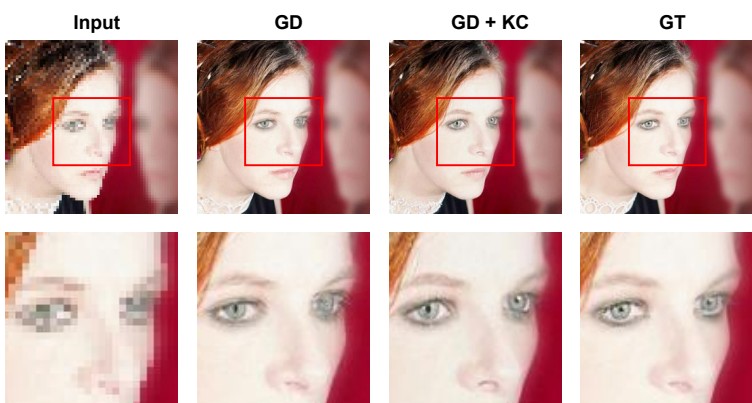

Figure 22: Qualitative comparison of with/without KC loss in guided diffusion (GD). The bottom image (with KC loss) has better eye and hair details (best viewed in color).

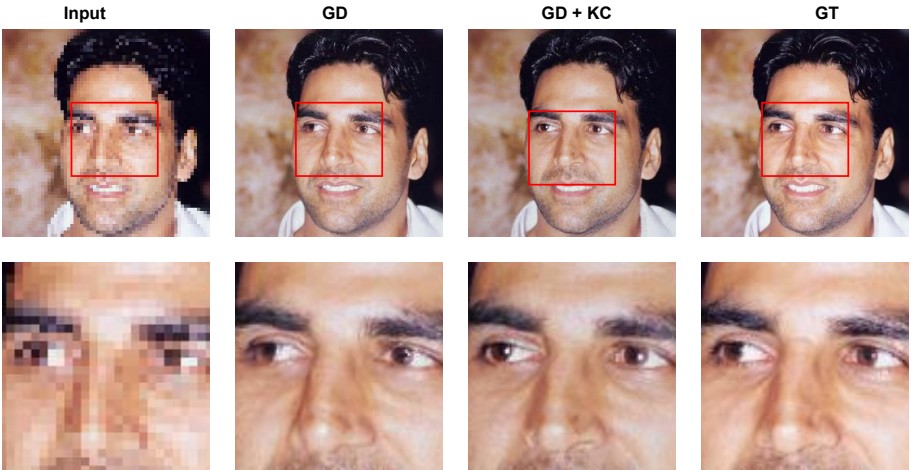

Figure 23: Qualitative comparison of with/without KC loss in guided diffusion (GD). The bottom image (with KC loss) has better eye details and skin smoothness (best viewed in color).

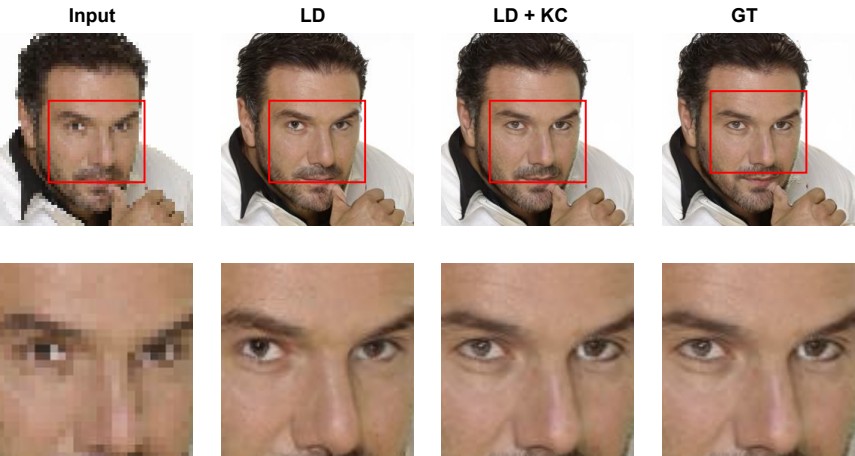

Figure 24: Qualitative comparison of with/without KC loss in Latent diffusion (LD). The bottom image (with KC loss) has higher similarity w.r.t the ground truth in terms of left eye and skin color (best viewed in color).

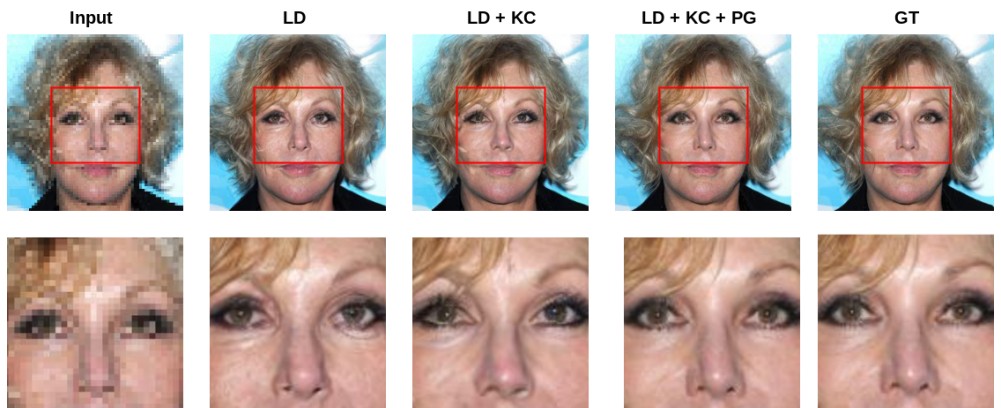

Figure 25: Qualitative comparison of with/without KC loss in Latent diffusion (LD). The bottom image (with KC loss) has higher similarity w.r.t the ground truth in terms of left eye and skin color (best viewed in color).

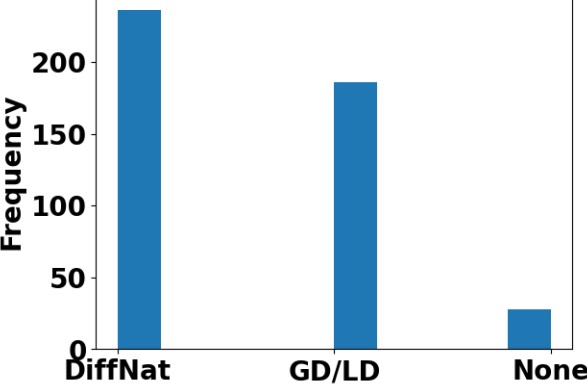

Figure 26: Human evaluation for image super-resolution task. DiffNat performs better than guided diffusion (GD), latent diffusion (LD) in user study as well.

Table 14: Comparison of image super-resolution (x8) task

| Method | Image quality | | | | |
|---|---|---|---|---|---|
| | FID score ↓ | PSNR ↑ | SSIM ↑ | LPIPS ↓ | MUSIQ score ↑ |
| GD Dhariwal & Nichol (2021) | 140.3 | 17.5 | 0.52 | 0.32 | 55.26 |
| GD + KC loss(Ours) | **125.5** | **18.7** | **0.56** | **0.27** | **57.33** |
| LD. Karras et al. (2022) | 103.2 | 18.7 | 0.59 | 0.25 | 58.62 |
| LD + KC loss(Ours) | **80.1** | **19.5** | **0.67** | **0.20** | **60.31** |

Table 15: Blind Face restoration

| Method | LPIPS↓ | FID↓ | IDS ↑ | PSNR ↑ | SSIM ↑ |
|--------|--------|------|-------|--------|--------|
| DifFace | 0.20 | 70.69 | 0.48 | 22.82 | 0.61 |
| RestoreFormer | 0.29 | 60.98 | 0.39 | 21.77 | 0.53 |
| IPC Suin et al. (2024) | 0.18 | 55.42 | 0.54 | 22.34 | 0.60 |
| IPC + KC | **0.15** | **43.21** | **0.61** | **24.19** | **0.64** |

| Input | IPC | IPC + KC | GT |
|-------|-----|----------|-----|

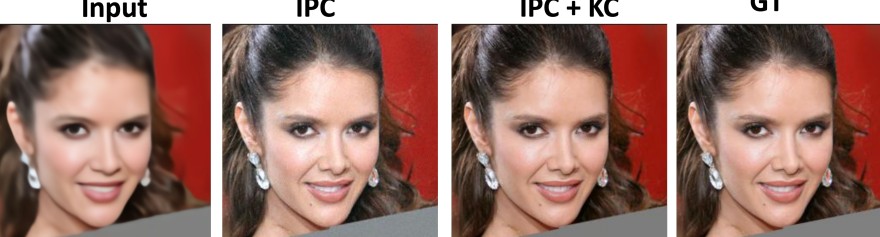

Figure 27: Restoration comparison with IPC

## L.1 BLIND FACE RESTORATION

For blind face restoration task Suin et al. (2024), we train on FFHQ dataset with and without KC loss on IPC baseline Suin et al. (2024), and evaluate on 3000 images on Celeb-A test set with a resolution of 256x256. Average LPIPS, FID, IDS, PSNR, SSIM are reported in Tab. 15. Qualitative results (Fig. 27) also verify that adding KC loss improves image quality.

## L.2 VIDEO EDITING

In text-based video editing Wu et al. (2023), input videos (e.g., goldfish swimming) need to be edited through text prompt (i.e., replace goldfish by shark, Fig. 28). Tune-a-video Wu et al. (2023) is a baseline method which perform this task in a one-shot finetuning way. We add our plug-and-play KC loss to the standard diffusion loss in Wu et al. (2023) pipeline, and observe that adding KC loss improves the editing quality. This is verified quantitatively, where we evaluate on the test set provided in Tune-A-video, i.e., 42 videos from DAVIS dataset wit 140 prompts. We measure the average CLIP frame similarity, CLIP text similarity and pick-score of the edited videos and report it in Tab. 16. Qualitatively we also verified this in Fig. 28.

Table 16: Video editing comparison

| Method | CLIP-Text ↑ | CLIP-Frame↑ | Pick score ↑ |
|--------|-------------|-------------|--------------|
| Tune-A-video Wu et al. (2023) | 26.40 | 94.83 | 29.33 |
| Tune-A-video + KC | **27.78** | **96.54** | **30.23** |

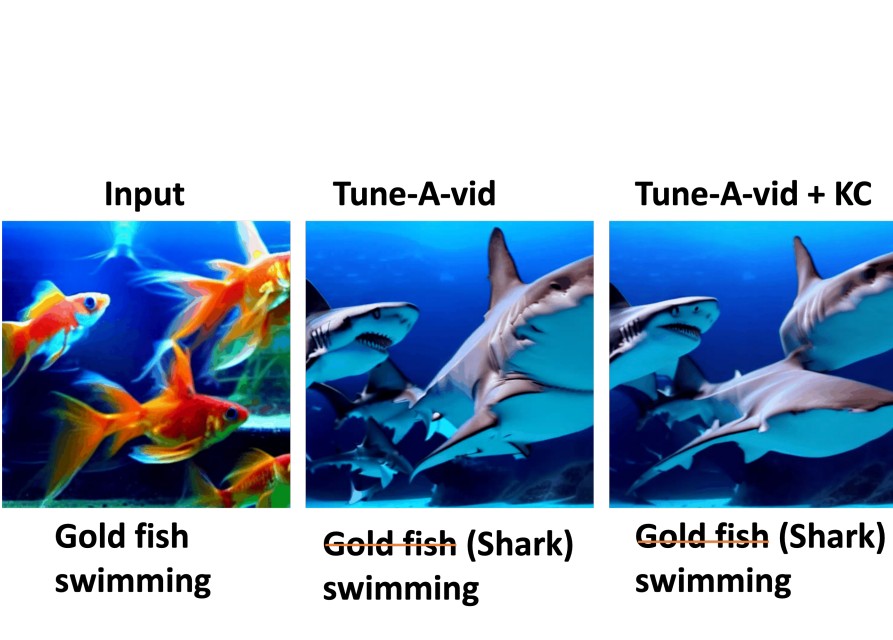

Figure 28: Video generation compare with Tune-A-video

