# OpenReview forum: "Exploiting the Kurtosis Concentration Property for Image quality improvement"
_ICLR.cc/2025/Conference — ICLR 2025 Conference Withdrawn Submission_

### Official Review · Reviewer_oDyv · 2024-10-21

**Soundness:** 3
**Presentation:** 1
**Contribution:** 2
**Rating:** 5
**Confidence:** 3

**Summary:**

This paper provides a clear summary and review of the Kurtosis Concentration Property, upon which this paper introduces the KC loss and PG strategy. Both methods are straightforward and easy to implement. Additionally, the paper tests its methods on multiple tasks and achieves promising results.

**Strengths:**

1. The paper reviews the research on the Kurtosis Concentration Property.

2. This paper proposes a concise KC loss and PG strategy.

3. The methods achieve promising results across diverse tasks and ablations experiments.

**Weaknesses:**

**1. Overclaiming Contribution Point 2**

This is the most concerning issue for me.

The second contribution point, which mentions "We provide insights on how reducing kurtosis improves image quality," is problematic. This contribution corresponds to section 3.1 of the paper, which revisits from the definition of the Kurtosis Concentration Property to its relationship to denoising. However, the content of this section has already been introduced in paper [1].

For instance, Definition 1 in section 3.1 corresponds to the first paragraph of section 3 in [1], Lemma 1 corresponds to Claim 1 in [1], and Lemma 2 corresponds to Equation 2  of [1]. Up to Lemma 2, this paper cites previous work including [1] reasonablely. However, when introducing Lemma 2, the paper fails to cite the previous work and introduces potentially misleading content. For example, in line 187, the paper states:
"Next, we establish the relation between the projection kurtosis of the noisy version of the image and the corresponding signal-to-noise ratio.", and introduce lemma 2.
However, in reality, Lemma 2 corresponds to Equation 2 in [1].

Therefore, the insights claimed by the authors are not original, leading to an overclaim of contribution. These insights are my favorite part of the paper, but it’s disappointing to discover that they are not entirely novel. This part should even be included in a Section named "Preliminaries".



**2. The Principle Behind the PG Strategy is Unclear**

The authors heavily promote the PG strategy as being condition-agnostic, but it seems to only aim at further improving image quality. It is unclear why this strategy is introduced. And there seems to be no clear motivation or explanation for why it achieves good results, making this section feel incomplete.



**3. The Paper's Presentation Needs Improvement**

3.1 **Please Use Vector Graphics:** Many of the figures, such as Figure 1, Figure 6, and others, are not vector graphics, making them blurry and lacking in detail even when zoomed in.

3.2 **Figures Do Not Highlight Method Improvements:** In comparative figures such as Figure 4, Figure 5, Figure 6, and Figure 9, the advantages in terms of texture detail and diversity can not be clearly observed. Figure 19 in the supplementary materials is clearer in this regard.

3.3 **The Paper’s Structure Could Be Further Optimized:** The structure could be rearranged to improve readability and flow, espefically for section 3.4.

---

[1] Zhang X, Lyu S. Using projection kurtosis concentration of natural images for blind noise covariance matrix estimation[C]//Proceedings of the IEEE Conference on Computer Vision and Pattern Recognition. 2014: 2870-2876.

**Questions:**

I greatly appreciate concise and effective work, especially when it focuses on modifying the loss function. The insights regarding the Kurtosis Concentration Property are the highlight of the paper for me, however, it's disappointing to find that they are not entirely novel. If these insights were the paper’s unique contribution, I would have accepted it without hesitation.

For specific Questions, please refer to the Weaknesses mentioned above in detail.

---

### Official Review · Reviewer_W7oS · 2024-10-28

**Soundness:** 2
**Presentation:** 3
**Contribution:** 2
**Rating:** 5
**Confidence:** 5

**Summary:**

The paper addresses the issue of unnatural artifacts and image quality in generative models. The authors propose a new loss function inspired by the Kurtosis Concentration (KC) property of natural images to tackle these challenges. Additionally, inspired by Classifier-Free Guidance (CFG), they introduce Perceptual Guidance (PG), which further enhances the overall quality of the generated images. The proposed method can be integrated into existing diffusion model pipelines, and comparisons across tasks demonstrate numerical improvements in FID, MUSIQ scores, and user evaluations.

**Strengths:**

1. The problem of artifacts, naturalness, and quality(perceptual) is important for the advancement of generative models.
2. KC loss is Lipschitz continuous and differentiable.
3. The method is easy to integrate into existing diffusion pipelines.
4. The paper is clearly written and easy to read.

**Weaknesses:**

1. With the advent of AIGC across all forms of professional and user-generated content (UGC), the KC property may not hold for modern databases. This limitation impacts the fundamental application and effectiveness of using a loss function with simple natural image statistical properties, such as KC loss. A detailed discussion on the KC property’s relevance for large-scale UGC-type databases should be discussed.

2. The "Constant-Kurtosis" property has long been used in perceptual quality models like DIVINE, NIQE, BRISQUE, and many others. The authors fail to discuss this, and using KC for (perceptual) quality is not novel. This limits the overall contribution of this paper.

3. The effectiveness of KC-loss is questionable:
(a) In Fig. 1, the highlighted artifact is small, and such artifacts are uncommon in generated images. This may simply be a poor choice of example, training issue, or poor prompting. Similar comments apply to Figs. 4, 5, 18, and 19.
(b) In Fig. 9, while the authors emphasize some corrected artifacts, more artifacts appear to be introduced. For example, in Fig. 9(b), the right eye is more deformed than in (a), and the region near the left ear shows increased deformation. Overall, there appears to be no significant perceptual quality improvement. Additionally, in Figs. 20, 22, 23, 24, 25, and 27, there is no perceptual quality difference, raising questions about the effectiveness of KC loss.
(c) Why are Fig. 4-DB and Fig. 10 ("A berry bowl with a mountain in the background") identical?
(d) If KC-loss improves the SNR, why do the PSNR values in Tables 3, 4, and 5 not show significant improvements compared to FID or MUSIQ?

4. In Fig. 6, there is very little difference between the GD and GD+KC images, and almost no difference between GD+KC and GD+KC+PG. This raises the question: is PG even improving performance? In Fig. 10, is the image shown with KC or KC+PG?

5. Although intuitive, but PG appears to be a forced novelty in the paper. It does not improve the KC-loss results, and very few experimental results are presented. Moreover, it requires two forward passes, making it time-consuming.

6. In Appendix F, the authors do not address a crucial question: if KC-loss only improves  the SNR, why does text-alignment performance decreases?

7. Minor but Necessary Improvements:
(a) Use high-resolution figures. For instance, In Fig. 1, it is difficult to see the artifacts and corresponding improvements. This applies to Fig. 10 as well.
(b) In Fig. 1, the authors do not clarify whether the figure is an overview of their method (DiffNat) including PG, or if it only shows KC loss. An overview should include both KC and PG.
(c) Line 851: Correct "strat" to "start."

**Questions:**

Please see the Weaknesses part.

---

### Official Review · Reviewer_LfTD · 2024-11-03

**Soundness:** 3
**Presentation:** 3
**Contribution:** 3
**Rating:** 6
**Confidence:** 4

**Summary:**

This paper introduces a novel loss function based on the projected kurtosis concentration property. The authors first connect kurtosis minimization to denoising and then present the KC loss. The proposed method is evaluated on three tasks, and it achieves promising performance while only leading to extra short raining time.

**Strengths:**

1.	The paper presents a novel loss for diffusion models by exploring the property of kurtosis concentration property.
2.	The papers also introduced an inference strategy that further improves image quality.
3.	The proposed methods lead to performance improvement in existing algorithms.

**Weaknesses:**

1.	Fig.1 is of low quality.
2.	Lemma 2 only considers the additive noise. Is the proposed method applicable to multiplicative noise? Or is this only used to connect the kurtosis concentration to diffusion models?
3.	The proposed loss will lead to additional training complexity.

**Questions:**

please see the weaknesses

---

### Official Review · Reviewer_i8bY · 2024-11-04

**Soundness:** 3
**Presentation:** 3
**Contribution:** 3
**Rating:** 3
**Confidence:** 4

**Summary:**

The paper proposes a kurtosis concentration (KC) loss to improve the image quality in diffusion models by utilizing the projected kurtosis concentration property of natural images. Meanwhile, they introduce a a condition-agnostic perceptual guidance strategy (PG) similar to classifier-free guidance during inference to further improve the image quality. The effectiveness of the method is validated in the tasks of text-guided image generation, unconditional image generation, and super-resolution  showing an improvement in image quality.

**Strengths:**

1. This paper provides a new perspective that utilizes the certain properties of natural images to improve the generation quality of generative models which can inspire further research on incorporating statistical properties of natural images to improve generative models.

2. This paper clearly explains the motivation and methodology, providing a theoretical justification for KC loss and PG.

**Weaknesses:**

1.  The proposed KC loss focuses primarily on visual quality, potentially compromising other aspects like diversity and semantic alignment.

2. Many figures in the paper (e.g., Fig. 1) appear blurry, and even when zoomed in I had difficulty seeing differences or artifacts.It is recommended that the authors replace them with vector graphics.

3. The FID score is a measure of the similarity of two distributions and typically requires 10k-30k images to be accurate, but Figure 1 appears to rely on only one image. Also, more relevant details about the Dreambooth experiment could be clarified.

4. The writing of the paper needs improvement; the tense is a bit confusing (for example, in the experiment tasks 1 and 2 used the past tense, while task 3 used the present tense), which affects readability.

5. This paper lacks specific validation regarding perceptual guidance (PC), specifically whether scale range and scale increase image quality consistently.

**Questions:**

During inference, how is the perceptual guidance (PC) combined with original CFG—by direct addition or another method?

---

### Note · Authors · 2024-11-15

I have read and agree with the venue's withdrawal policy on behalf of myself and my co-authors.